# Predictive Data Selection: The Data That Predicts Is the Data That Teaches

**Kashun Shum** [* 1]  **Yuzhen Huang** [* 1]  **Hongjian Zou** [2]  **Qi Ding** [2]  **Yixuan Liao** [2]  **Xiaoxin Chen** [2]  **Qian Liu**
**Junxian He** [1]

## Abstract

Language model pretraining involves training on extensive corpora, where data quality plays a pivotal role. In this work, we aim to directly estimate the contribution of data during pretraining and select pretraining data in an efficient manner. Specifically, we draw inspiration from recent findings showing that compression efficiency (i.e., normalized loss) of diverse models on certain text correlates strongly with their downstream performance, when the text domain aligns with the downstream benchmarks (Huang et al., 2024). Building on this observation, we hypothesize that *data on which model losses are predictive of downstream abilities also contribute effectively to learning*, which shares similar intuition with Thrush et al. (2024). To leverage this insight, we introduce *predictive data selection* (PRESELECT), a lightweight and efficient data selection method that requires training and deploying only a fastText-based scorer. Through comprehensive experiments with 1B and 3B parameter models, we demonstrate that models trained on 30B tokens selected with PRESE-LECT surpass the performance of the vanilla baseline trained on 300B tokens, achieving a 10x reduction in compute requirements. Furthermore, PRESELECT significantly outperforms other competitive data selection baselines, such as DCLM and FineWeb-Edu on a scale of 3B models trained on 100B tokens. We open-source our trained data selection scorer along with the curated datasets at https://github.com/hkust-nlp/PreSelect.

## 1. Introduction

Large language model (LLM) pre-training typically requires training on a huge data source such as web crawl data

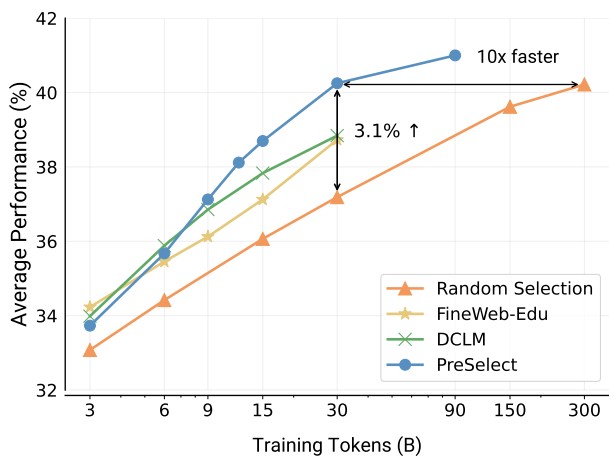

Figure 1: PRESELECT outperforms random selection by an average of 3.1% accuracy across 15 downstream benchmarks and achieves 10x reduction in compute requirements on the scale of 1B models using RefinedWeb corpus.

where the existence of low-quality data leads to slow scaling law (Kaplan et al., 2020; Chowdhery et al., 2023). Given the growing available token budget and limited training budget nowadays, data selection has become a standard step in the pre-training stage. This step has played a critical role in enhancing data quality and optimizing training efficiency during the development of various LLMs (Touvron et al., 2023b; DeepSeek-AI, 2024). While previous practices often rely on human heuristics such as pre-defined rules and domain classification to filter data (Penedo et al., 2024b; Li et al., 2024a), in this work we aim to directly identify data that could contribute effectively to learning various abilities.

We are inspired by the belief that "compression represents intelligence" and the corresponding empirical finding recently (Huang et al., 2024), that the compression efficiency (i.e., the normalized loss)[1] of a series of models on certain raw text, strongly correlates with their downstream benchmark performance, when the text data aligns with the benchmark domains. For example, Huang et al. (2024) observed that model losses on raw GitHub code exhibit an

---

[*]Equal contribution  [1]HKUST [2]Vivo AI Lab. Correspondence to: Kashun Shum <ksshumab@connect.ust.hk>, Junxian He <junxianh@cse.ust.hk>.

*Proceedings of the 42nd International Conference on Machine Learning*, Vancouver, Canada. PMLR 267, 2025. Copyright 2025 by the author(s).

---

[1]For a given model, the lossless compression efficiency, such as bits per character on certain data, is equal to the normalized loss on that data (Deletang et al., 2024; Huang et al., 2024). See §2.1 for further details.

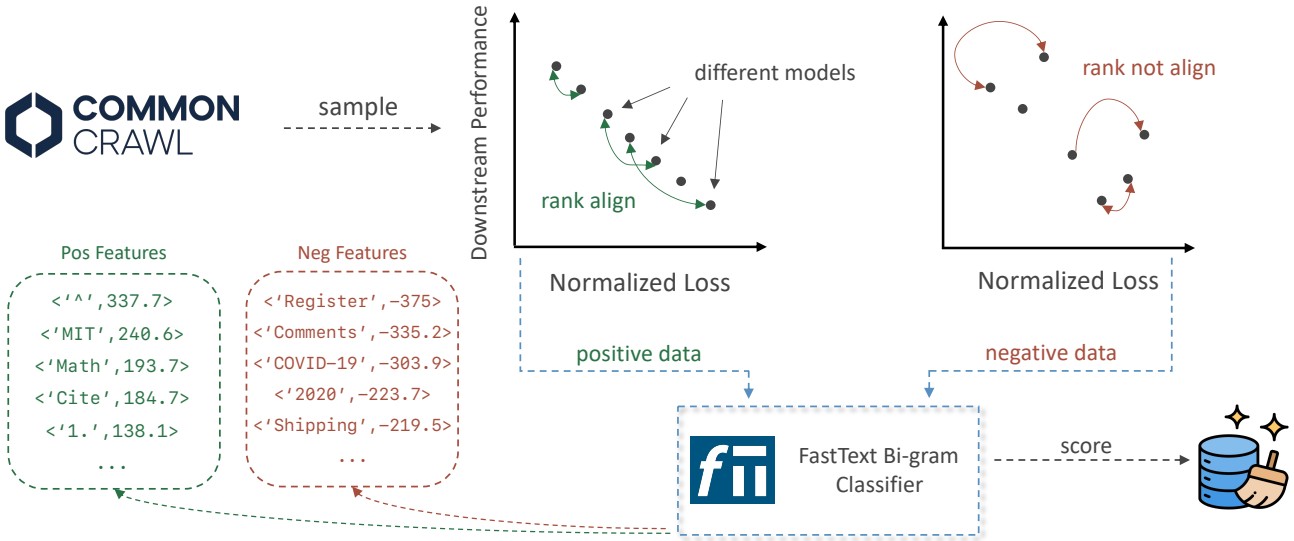

Figure 2: The overall framework of PRESELECT. It first samples a subset of pre-training data for computing the predictive strength score of each document (§2.2). Then a fastText-based scorer is trained based on the identified positive data and negative data. Finally the fastText-based scorer is trained to enable scalable data selection.

almost linear correlation with performance on various coding tasks, whereas model losses on Common Crawl data are more indicative of performance on knowledge-intensive tasks. These insights suggest that compression efficiency on certain data is more reflective of model's intelligence than that on other data. Motivated by this, we hypothesize that *data on which compression more effectively represents intelligence also facilitates the learning of that respective intelligence more effectively.* A similar hypothesis "LLM losses on many pretraining texts are correlated with downstream benchmark performance" was proposed by Thrush et al. (2024) previously, but the empirical validation was limited and the effectiveness remained inconclusive.

Building on this hypothesis, we first define the *predictive strength* of a given text to quantify how well the compression efficiency on it correlates with the models' ability, then, we introduce *predicative data selection* (PRESELECT), a data selection method based on the data's predictive strength. Different from previous works that try to quantify the pretraining data influence requiring controlled training (Yu et al., 2024; Engstrom et al., 2024), our approach is extremely lightweight, and only utilizes open pretrained models in the wild without training any deep models in the process. After collecting a small seed dataset, a fastText (Joulin et al., 2017) is trained to enable scalable data selection.

While we are motivated independently by Huang et al. (2024), our approach shares similar intuition with Perplexity Correlation (Thrush et al., 2024) and can be viewed under the framework of Perplexity Correlation, which is the first to explore domain-level correlation-based methods in data selection. However, this work differs from Thrush et al.

(2024) in several key aspects: (1) While they compute perplexity correlations for data domains (e.g., wikipedia.com), our method operates at a much finer granularity, directly operating on and selecting individual documents.[2] As we will demonstrate in §3.4 and §4.3, this finer granularity leads to significant differences in the selected data and, consequently, in the final results; (2) Perplexity correlation suggested using a large number of (e.g., around 90) open-sourced models to approximate the correlation. However, we argue that using many open-sourced LLMs brings potential evaluation noises when dealing with different families of the models and causes computation inefficiency. Thus in our design, we employ 6 Llama models only during data selection. (3) They performed very limited experiments with a 160M-size model and compared with several baselines on limited benchmarks, while we train models with up to 3B parameters on up to 100B tokens and evaluate on 17 diverse benchmarks, a more realistic scale in a typical pretraining setting. The updated, concurrent version of Thrush et al. (2024) further added 1B-scale experiments assesed on comprehensive benchmarks, where the results indicate good performance on raw data pools but no improvement on high-quality, well-processes data pools – an observation distinct from our empirical results. PRESELECT targets a well de-duplicated, high-quality pre-training corpus – which we believe represents a realistic setting – and shows the effectiveness of selecting data from high-quality data pools

---

[2]We note that Thrush et al. (2024) preregistered document-level experiments as future tasks, which were not present as we released the first version of this paper. The updated version of Perplexity Correlation added document-level results concurrent to this work, but the final results are very different as we will dicuss in §2.3.

in §3.4.

Experimental results show that PRESELECT naturally selects high-quality data, achieving an average of 5.3% absolute improvement over random selection at 1B scale and outperforms current state-of-the-art data selection methods by 2.2% absolutely on tasks across understanding, knowledge, math, and code. PRESELECT also show its adaptability to different model architectures (Llama (Touvron et al., 2023a) and Pythia (Biderman et al., 2023)), as well as different pre-training corpora across different model sizes from 400M to 3B. Compared to training without data selection, PRESELECT reduces training steps by up to 10x as shown in Figure 1. At the meanwhile, by eliminating data with negative effect, PRESELECT achieves superior performance to training on the entire corpora.

## 2. Data Selection with Predictive Strength

### 2.1. Preliminaries – Relationship Between Compression and Intelligence

Language modeling can be considered a form of lossless compression (Deletang et al., 2024). From this regard, given a sequence of text tokens $\{x_i\}_{i=1}^n$ that contain $T$ characters, the average language model loss per character $\sum_{i=1}^n -\log_2 p_{model}(x_i|x_{1:i-1})/T$ represents the minimal number of bits required to encode one character on average. Therefore, the normalized language loss on given text $x$ represents how efficiently a model can compress $x$ in a lossless manner. Recently, Huang et al. (2024) empirically verified that the average downstream task scores across various diverse models strongly correlate with their normalized losses on certain raw text data, even though these models are pretrained on distinct data with different tokenizers. For example, they found that normalized losses on Common Crawl correlated strongly with the models' performance on knowledge-intensive tasks. A similar level of correlation was also observed between losses on GitHub code files and performance on code generation tasks. However, such strong correlations are not universally present – for instance, the losses on Common Crawl show a weak correlation with tasks that do not rely on factual knowledge.

This insight underscores that not all data are equally reflective of models' abilities through the losses on them. Inspired by this observation, we hypothesize that data on which compression or losses more effectively represent models' ability is better suited for learning that respective ability. This perspective forms the foundation of our proposed approach, which we introduce next.

### 2.2. Data's Predictive Strength

Here, we aim to quantify how effectively the losses on a given dataset represent the model's capabilities.

While Huang et al. (2024) use the Pearson correlation coefficient to measure the relationship between losses and the model's downstream performance, we take a simplified approach by examining how well the rankings of the models based on losses align with their rankings on downstream benchmark scores. Specifically, assume we have a series of $N$ open-weight pre-trained language models $\{M_1, M_2, ..., M_N\}$ alongside their averaged benchmark score $\{S_1 < S_2 < ... < S_N\}$. Given any document in the pre-training corpus $d \in D$, we can compute each model's normalized losses per character on $d$, and obtain $\{C_1, C_2, ..., C_N\}$. We adopt a simple matching score as the correlation between the ranks of normalized losses and ranks of benchmark scores:

$$\mathbf{S} = \sum_{1 \le i < N} \sum_{i < j \le N} \mathbb{I}\{C_i > C_j\}/Z, \tag{1}$$

where $Z = \frac{N^2-N}{2}$ is the normalization factor to ensure $S \in [0, 1]$. Intuitively, Eq. 1 evaluates whether the ranking of losses for each pair of models is inversely aligned with their task score rankings (since lower values are better for losses). The score $\mathbf{S}$ increases when the rankings are inversely aligned, indicating that the losses on this dataset effectively reflect the models' rankings on downstream scores. A score of 1 implies that the losses on the document can perfectly predict the models' downstream performance rankings. Accordingly, we refer to the score in Eq. 1 as the document's *predictive strength*.

Beyond considerations of simplicity, we argue that this ranking-based correlation is more robust to noise compared to the Pearson correlation used by Huang et al. (2024) in document-level calculation. As in prior, Perplexity Correlation (Thrush et al., 2024) has explored ranking-based metric to estimate correlation coefficient in domain-level. Loss computation on a single document can be sensitive to noisy or anomalous text, especially when the document is short. In such cases, numerical correlation estimation such as Pearson correlation, which reacts sharply to minor variations in loss values, may not be a suitable metric. This instability can, to some extent, be mitigated by grouping documents first and selecting based on these groups. However, such a grouping step introduces additional human heuristics into the process that we think may be suboptimal, as we will verify empirically in §3.4.

We propose to select documents with high predictive strength scores, hypothesizing that the data that can be used to predict the model's ability is also the data that contributes to learning it. Different from selection strategies based on pre-defined rules such as FineWeb-Edu (Penedo et al., 2024b) which prioritizes education-related documents, or DCLM (Li et al., 2024a) which selects referring to supervised-finetuning data, our method is more principled and bypasses strong human heuristics.

## 2.3. The Overall Framework

Our core hypothesis is that the data on which the losses can help predict the performance well is the data that can contribute to training effectively, thus the documents' predictive strength scores as defined in Eq. 1 are our metric to select data. However, computing Eq. 1 for every document is expensive since it requires computing losses of several models on all the data, thus we follow Li et al. (2024a) and only select a small seed set of documents that are later used to train a fastText classifier (Joulin et al., 2017). As shown in Figure 2, we first randomly sample a small subset of data from the pre-training corpus, which is kept separate from the data used for later training, to compute the normalized losses. Specifically, we pick the 3,000 most frequent domains (e.g., wikipedia.org) in the corpus and randomly sample 300 examples for each domain for a wide coverage over the pre-training corpora, thus obtaining 900K samples in total. Then we choose the models from the Llama 1 and 2 series (Touvron et al., 2023a;b) ranging from 7B to 65B parameters – in total 6 models are used to compute the predictive strength for the data. We do not choose various models from different families to compute the predictive strength on purpose – in our preliminary experiments, we found that models from various families introduce significant evaluation noises, for example, they are sensitive to prompts and the prompt that is good for one model may not be suitable for another. This issue is particularly salient given that we are evaluating base models that are generally more sensitive to prompts than instruction-tuned models. Such noises would dramatically affect the computation of predictive strength, and thus we found that data selected this way could not outperform a random data selection baseline in our initial trials. This design differs from Thrush et al. (2024) which employs diverse models to estimate perplexity correlation. Such distinction probably explains why perplexity correlation fails to improve random data selection baselines significantly on a high-quality, pre-filtered data pool, as shown in the concurrent version of their paper. We run inference of these models on the 900K documents to obtain the loss ranks of these models on each document respectively. Next, we obtain the downstream score ranks by directly using the averaged scores on a diverse set of 12 benchmarks from Huang et al. (2024). We also explored how performance change when choosing one specific task (e.g. HellaSwag) as the target that elicits a different ranking to averaged score in Appendix A.7.2.

Once we compute the predictive strength scores for all the documents, we identify the positive and negative samples from them to train a fastText classifier. Concretely, we select the documents with the highest predictive strength scores as the positive examples, and those with the lowest predictive strength scores as the negative ones. We select around 200K positive and 200K negative examples to train a fastText

scorer. Notably, our approach requires only the fastText scorer at deployment, making it highly accessible and scalable to select data from a large corpus. We refer to our data selection method as PRESELECT, short for *predictive data selection*.

## 3. Experiments

In this section, we first introduce our pre-training corpus (§3.1). Then various pre-training data selection baselines we compared (§3.2) and evaluation settings (§3.3) are presented. In the last part, we will demonstrate our experimental results. Full implementation details and evaluation details are illustrated in Appendix B and Appendix C.

### 3.1. Pre-training Corpus and Models

For a fair comparison and ease of processing, we follow and directly use the large data pool created in Li et al. (2024a). Concretely, this data pool utilized a version of RefinedWeb that undergoes processing through *resiliparse* text extraction (Bevendorff et al., 2018), RefinedWeb's heuristic filtering (Penedo et al., 2024d) and deduplication using bloom filters (Soldaini et al., 2024), solely filtered from Common Crawl. This large data pool has over 20 trillion tokens in total, which will serve as the source for us to sample smaller data pools to conduct data selection experiments in various scales. We refer to the pool as RefinedWeb in the following sections.

Following the suggested setting in Li et al. (2024a), we randomly sample 80 billion, 300 billion, and 1 trillion tokens as the data selection pool for 400M, 1B, and 3B model training respectively and select 10% of the data for training unless otherwise specified, which corresponds to the Chinchilla optimal training data size (Hoffmann et al., 2024). For the models, we use a Llama architecture (Touvron et al., 2023a) unless otherwise specified. We also apply PRESELECT on the C4 dataset to show the effectiveness of our method on different corpora and to compare with more data selection baselines.

### 3.2. Baselines

To make a fair comparison, we keep the data pool and training setting of baselines the same as PRESELECT, such as model architecture and training hyper-parameters, while only the data selection part is different. We consider the following baselines: (1) *Random*, which randomly selects the documents; (2) *PPL filtering*, which keeps low-perplexity documents following CCNet (Wenzek et al., 2020); (3) *FineWeb-Edu* (Penedo et al., 2024b), which scores the education level of documents with an LLM and then train a small scorer to identify and select educational contents (Penedo et al., 2024b); (4) *PPL Correlation (DD)* (Thrush et al.,

2024), which utilizes domain-level correlation estimation and performs domain-level selection. (5) *PPL Correlation (DP) (Thrush et al., 2024), which utilizes domain-level correlation estimation and trains a fasttext based on domain-level positive/negative samples to perform page-level selection*. As discussed before in §1, these two baselines are the most relevant to PRESELECT while our approach does not rely on human heuristics to group documents and operates at the document level directly; and (6) *DCLM (Li et al., 2024a)*, which is the state-of-the-art pretraining data selection method that trains the fastText scorer using supervised fine-tuning (SFT) data as the positive data. For FineWeb-Edu and DCLM, we directly use their released BERT and fastText scorer to select documents in our pool. Due to the high cost of running all the baselines across all settings, we mainly include all of them in our setting of 1B model training for 30B tokens. (7) *ScalingFilter (Li et al., 2024b)*, which uses the perplexity difference between a large model and a small model as the metric to select data, which we compared and discussed how intermediate level models actually help in Appendix A.7.1. In other experiment scales, we only compare to DCLM which is superior to other baselines. We provide more details about the model architecture and training hyper-parameters in Appendix B.3 and Appendix B.4 respectively.

In an ablation experiment to validate our approach on the C4 data pool, we also compare with more baselines such as DSIR (Xie et al., 2023) which selects data by importance resampling, DsDm (Engstrom et al., 2024) which uses datamodels to approximate the relationship between data subset and benchmark, QuRating (Wettig et al., 2024) which aims to capture human intuitions and MATES (Yu et al., 2024) which selects data dynamically based on data influence model. These baseline numbers are directly taken from Yu et al. (2024). There are other widely used methods aimed at improving data quality such as data refinement (Zhou et al., 2024). We do not include comparison with such methods as data refinement is orthogonal to data selection.

### 3.3. Evaluation Settings

We evaluate the pre-trained base models on 17 tasks across different general domains which include: MMLU (Hendrycks et al., 2020), Arc-Easy, Arc-Challenge (Clark et al., 2018), HellaSwag (Zellers et al., 2019), PIQA (Bisk et al., 2020), SIQA (Sap et al., 2019), SciQ (Welbl et al., 2017), RTE (Dagan et al., 2005), BBH (Suzgun et al., 2022), LAMBADA (Paperno et al., 2016), OpenBookQA (Mihaylov et al., 2018), RACE-Middle, RACE-High (Lai et al., 2017), MultiRC (Khashabi et al., 2018), WinoGrande (Sakaguchi et al., 2021). We also aim to evaluate the Math and Code domains, however, we found that our models of these scales can only yield negligible performance on widely used problem-solving

benchmarks such as GSM8K (Cobbe et al., 2021) and HumanEval (Chen et al., 2021). Therefore, we follow the advice of Huang et al. (2024) and report the bits per character (BPC) on Math-related and GitHub raw texts, where we directly use their released Math and Code evaluation corpora. For experiments under C4 corpus, we follow MATES's setting and evaluate the zero-shot performance on Arc-Easy, Arc-Challenge, SciQ, LogiQA (Liu et al., 2020), OpenBookQA, HellaSwag, PIQA and WinoGrande The full evaluation details are illustrated in Appendix C.

### 3.4. Main Results

**Comparison with Baselines** As shown in Table 1 top part, we verify the effectiveness of our method by conducting initial experiments on 400M models. PRESELECT demonstrates remarkable performance, with an average absolute improvement of 2.8% over the random selection and 20% gains in Math and Code raw text BPC, which shows a promising trend. Scaling to the 1B model, as shown in the middle part of Table 1, the results show that PRESELECT has the best performance on most tasks, outperforming random selection with an averaged absolute improvement of 5.3% on 7 representative tasks, including significant boosts such as 8.8% on Arc-Easy, 8.4% on BBH and 6.7% on SciQ. In greater detail, as shown in the full evaluation table C.2, on an average of 15 benchmarks, we outperform random selection by 3.1% absolute improvements. FineWeb-Edu shows notable improvements on exam-related tasks such as ARC, however, the improvement on other abilities such as understanding is relatively marginal. More importantly, PRESELECT outperforms the strongest data selection method DCLM by over 2%, achieving better performance on most tasks. Similar to the 400M scale, PRESELECT consistently shows significant improvements in Math and Code domain with an averaged improvement of 19% and 18% on raw text BPC.

Compared with Perplexity Correlation (DD)which selects domains based on the correlation of each domain, it only achieves relatively marginal improvements over random selection and is significantly underperformed PRESELECT by 4.8%. As we will further discuss in § 4.3, the removal of many domains leads to a significantly decreased diversity, thus, some tasks such as HellaSwag would even have a significant drop. In addition, we compare with Perplexity Correlation (DP) which perform page-level selection using fastText trained with domain-level positive/negative samples. We see that though it has 1.2% improvements over random baseline, it still underperformed PRESELECT by 4.1%, showing the effectiveness of using example-level correlation estimation. This difference is because by operating at the domain level, it may inadvertently include low-quality data from the high-correlation domains while overlooking high-quality data from unselected domains when building

Table 1: The experimental results of different data selection baselines ranges from 400M to 3B on the RefinedWeb data pool. For ease of space, we only show a subset of representative tasks here from each domain, while we include additional results in Appendix C.2. **Bold** denotes the best. Gray is provided for reference only, as it is trained on more tokens. Math and Code are measured by bits per character(BPC) while others use accuracy.

| Method | Tokens | ARC-E | ARC-C | MMLU | LAMBADA | RACE | SciQ | BBH | Avg. | Math (↓) | Code (↓) |
|---|---|---|---|---|---|---|---|---|---|---|---|
| 400 Model with 10% selection threshold | | | | | | | | | | | |
| Random | | 33.5 | 19.3 | 25.9 | 13.2 | 21.5 | 56.2 | 2.2 | 24.2 | 1.224 | 1.111 |
| DCLM | 8B | 38.6 | 19.7 | **26.1** | 14.3 | 21.7 | 60.2 | 3.1 | 25.7 | 1.031 | 0.929 |
| PRESELECT(ours) | | **41.6** | **22.7** | 26.0 | **14.8** | **23.1** | **61.2** | **3.7** | **27.0** | **0.995** | **0.885** |
| 1B Model with 10% selection threshold | | | | | | | | | | | |
| Random | 300B | 42.2 | 27.8 | 24.5 | 27.6 | 22.3 | 70.9 | 12.8 | 31.3 | 0.892 | 0.804 |
| Random | | 39.2 | 24.4 | 26.0 | 19.0 | 21.9 | 64.8 | 7.8 | 28.1 | 1.023 | 0.901 |
| PPL Filtering | | 42.5 | 24.6 | 25.8 | 18.8 | 22.6 | 67.5 | 8.5 | 29.1 | 0.957 | 0.853 |
| FineWeb-Edu | | **48.3** | 26.1 | 26.0 | 18.2 | 24.4 | 69.0 | 12.8 | 31.1 | 0.906 | 0.816 |
| PPL Correlation (DD) | 30B | 39.7 | 23.7 | 26.1 | 20.7 | 22.8 | 63.7 | 9.5 | 28.6 | 0.980 | 0.919 |
| PPL Correlation (DP) | | 44.2 | 24.6 | 25.2 | 19.9 | 22.9 | 65.7 | 8.8 | 29.3 | 0.982 | 0.833 |
| DCLM | | 45.2 | 24.8 | **26.3** | 22.2 | 24.3 | 70.0 | 12.6 | 31.2 | 0.857 | 0.773 |
| PRESELECT(ours) | | 48.0 | **26.8** | 26.0 | **23.5** | **27.7** | **71.5** | **16.2** | **33.4** | **0.830** | **0.744** |
| 1B Model with 30% selection threshold | | | | | | | | | | | |
| DCLM | 90B | 47.6 | **27.5** | **26.3** | 26.2 | 22.7 | 74.7 | 13.3 | 32.6 | 0.847 | 0.771 |
| PRESELECT(ours) | 90B | **49.2** | **27.5** | 26.0 | **27.0** | **25.0** | **75.4** | **17.2** | **34.0** | **0.831** | **0.757** |
| 3B Model with 10% selection threshold | | | | | | | | | | | |
| Random | | 51.2 | 29.2 | 24.8 | 33.2 | 22.5 | 79.5 | 15.3 | 34.7 | 0.818 | 0.726 |
| DCLM | 100B | 55.7 | 31.2 | 25.3 | 35.1 | **26.0** | 82.5 | 20.5 | 37.8 | 0.712 | 0.664 |
| PRESELECT(ours) | | **61.2** | **31.9** | **26.2** | **36.1** | 25.8 | **85.6** | **23.3** | **39.5** | **0.694** | **0.648** |

the fastText classifier.

In the scale of the 3B model trained on 100B tokens, as shown in the bottom part of Table 1, the results indicate that PRESELECT consistently achieves the best performance across almost all representative tasks compared to other baselines, demonstrating the scalability and effectiveness of PRESELECT. Specifically, similar to the 1B setting, PRESELECT outperforms random selection by a significant margin in tasks such as Arc-Easy and BBH, with 10% and 8% absolute increase respectively. Moreover, PRESELECT still achieves the best performance on Math and Code.

**Efficient Training with Data Selection**  Data selection as a standard approach to save computation, it is also common to see how many training steps it can save compared to training without data selection. We directly train the 1B model from scratch with full data size 300B (10x selected data). It is surprising that PRESELECT with 30B tokens shows superior results to the model trained with 300B tokens, indicating a **10x** reduction in computation requirement. Also, by comparing DCLM-selected data and PRESELECT-selected data at 30% selection ratio (90B tokens), PRESELECT consistently show a better performance than DCLM. In addition, by comparing DCLM trained with 90B and tokens and PRESELECT trained with 30B tokens, DCLM needs 3x more training tokens to show comparable results to PRESELECT, indicating greater efficiency of PRESELECT.

### 3.5. Using C4 as the Data Pool

Next, we perform experiments on a different setting to further validate PRESELECT with different data pools and model architectures. Specifically, following the MATES's (Yu et al., 2024) setting, we train a 410M and 1B Pythia-architecture (Biderman et al., 2023) model with 25B tokens as shown in Table 2. We directly include the baseline results from Yu et al. (2024). PRESELECT demonstrates an absolute averaged improvement of 2.2% and 2.1% over random selection at 410M and 1B model respectively, as well as notable improvements over all other data selection baselines, showing its applicability across different pretraining corpora and model architectures.

## 4. Analysis

In this section, we analyze the data selected by various data selection methods to gain insights into the characteristics of the selected data.

### 4.1. Positive/Negative Data Analysis

As discussed in §2.3, PRESELECT defines positive data as those with high predictive strength scores. What data are selected as positive data and negative data respectively? We observed that literature-related domains contain a higher percentage of positive data such as *literotica* and *ukessays*,

Table 2: The zero-shot performance comparison between data selection baselines and PRESELECT on C4 with Pythia model architecture. All reference results of baselines are obtained directly from (Yu et al., 2024). **Bold** denotes the best.

| Method | SciQ | ARC-E | ARC-C | LogiQA | OBQA | HellaSwag | PIQA | WinoGrande | Average |
|---|---|---|---|---|---|---|---|---|---|
| | | | | 410M Model trained with 25B tokens | | | | | |
| Random | 64.1 | 40.2 | 25.6 | 24.7 | 29.4 | 39.7 | 67.1 | 50.6 | 42.7 |
| DSIR | 63.1 | 39.9 | 23.8 | **27.0** | 28.4 | 39.6 | 66.8 | 51.5 | 42.5 |
| DsDm | 65.4 | 41.7 | 24.7 | 27.5 | 29.0 | 40.3 | 68.1 | 50.1 | 43.4 |
| QuRating | 64.8 | 42.0 | 25.4 | 25.3 | 30.2 | 40.7 | 67.5 | 52.1 | 43.5 |
| MATES | 66.0 | 41.8 | 25.0 | 25.7 | 30.8 | **41.0** | **68.7** | 52.7 | 44.0 |
| PRESELECT(ours) | **67.9** | **47.0** | **26.2** | 26.4 | **31.0** | 40.5 | 67.3 | **52.9** | **44.9** |
| | | | | 1B Model trained with 25B tokens | | | | | |
| Random | 65.8 | 43.7 | 25.6 | 27.5 | 31.8 | 43.8 | 68.9 | 50.7 | 44.7 |
| DSIR | 65.8 | 42.6 | 24.7 | 28.7 | 29.2 | 44.2 | 68.3 | **53.2** | 44.6 |
| DsDm | 68.2 | 45.0 | 26.5 | 26.6 | 29.4 | 44.8 | 68.9 | 51.9 | 45.2 |
| QuRating | 67.1 | 45.5 | 25.6 | 26.9 | 29.8 | 45.2 | **70.2** | 51.6 | 45.2 |
| MATES | 67.3 | 44.9 | 25.9 | 28.7 | 32.2 | **45.3** | 69.5 | 52.4 | 45.8 |
| PRESELECT(ours) | **69.5** | **50.4** | **27.4** | **28.9** | **32.4** | 44.8 | 68.7 | 52.6 | **46.8** |

Table 3: The Top-15 fastText learned fearures with corresponding influence scores of different methods. Positive and Negative are defined according to the magnitude of the influence score.

| Method | Top-15 fastText Features with Influence Scores |
|---|---|
| DCLM (pos) | (""', 968.194), (Why, 671.558), (Answer:, 620.404), (Additionally,, 599.6), (assistant., 567.250), (Edit:, 565.787), (Generate, 537.11), (answer., 502.883), (Given, 474.243), (A:, 457.440), (Question:, 447.825), (Write, 440.69), (task., 423.878), (ELI5, 419.001), (AI, 403.29) |
| DCLM (neg) | (–, -479.818), (Comment, -466.0545), (Posted, -450.19833), (Post, -442.57623), ([. . .], -386.1609), (Comments, -377.41232), (—, -344.9523), (About, -340.93152), (. . ., -321.9254), ("The, -321.84586), (Re:, -310.0053), (More, -297.77707), (», -294.49173), (See, -289.7501), (2012, -278.6206) |
| PPL Correlation (DP)(pos) | (Archives, 347.5), (Navigation, 299.7), (Commenting, 288.5), (Section, 280.3), (E-mail, 276.1), (Played:, 263.9), (HB, 249.7), (Lyrics, 240.4), (Received, 231.4), (Gamercard, 226.7), (+0000, 226.6), (Discussion:, 224.3), (Sleeps, 219.0), (Playing:, 216.9), (Forgot, 216.3) |
| PPL Correlation (DP)(pos) | (Login, -459.9), ((permalink, -367.6), ((IANS, -358.0), (Images, -344.0), (macrumors, -343.3), ((credit:, -320.8), (Shipping, -314.9), (Subscription, -293.5), (Defines, -291.0), (Number:, -286.5), (Lainey, -279.2), (Maine, -278.0), (Tutors, -277.6), (Nearby, -269.7), (By:, -269.5) |
| PRESELECT (pos) | (ˆ, 337.7), (Likes, 241.387), (sharelimprove, 241.28), (MIT, 240.614), (–\xa0, 240.006), (Retrieved, 232.997), (Received, 214.872), (Photo:, 205.808), (Azure, 200.453), (Tolkien, 195.01), (Math, 193.741), (answer, 187.706), (Contributed, 185.518), (Cite, 184.729), (MATLAB, 176.991) |
| PRESELECT (neg) | (Register, -375.421), (Details, -335.247), (Comments, -320.821), (Updated:, -310.755), (COVID-19, -303.878), (tho, -281.664), (Print, -271.024), (Profile, -262.205), (Leave, -258.85), (Published:, -235.235), (Product, -228.646), (coronavirus, -227.227), (2020, -223.704), (By:, -222.939), (Shipping, -219.518) |

followed by knowledge related domains such as *wikipedia*, *stackexchange* and *wikihow*. While for negative data, the top domains vary widely, encompassing reference materials, commerce sites, genealogy platforms, and others such as *abbreviations.com, onegreatfamily.com, citysearch.com...*

### 4.2. FastText Feature Contribution

After selecting the small set of representative positive and negative data, PRESELECT trains a fastText-based scorer so that the method can be adapted to the large corpus. Because of the simple architecture of fastText classifier which con-

tains two linear transformations - input matrix and output matrix, where the input matrix maps all uni-gram features to the hidden dimension and the output matrix maps hidden dimension to final classification dimension, and in our case, is 2, we are able to track the influence of individual uni-gram on final prediction score (the detailed calculation can be found in §A.3). This helps understand what uni-gram patterns the fastText classifier looks for and potentially explains the word-level characteristic of good/bad data.

As shown in Table 3, for DCLM, the baseline that also uses a fastText classifier, the uni-gram features with high influ-

Table 4: The selected data distribution (RefinedWeb) over source domains of different data selection baselines. The top-15 domains are listed for comparison and the density are counted based on character percentage.

| Method | Top-15 Domains (Percentage Density) |
|---|---|
| Random | en.wikipedia.org (0.36), issuu.com (0.15), ufdc.ufl.edu (0.13), www.theguardian.com (0.12), www.fanfiction.net (0.12), www.nifty.org (0.11), www.beeradvocate.com (0.10), openjurist.org (0.09), www.nytimes.com (0.08), bleacherreport.com (0.08), www.barnesandnoble.com (0.06), www.washingtonpost.com (0.06), www.huffingtonpost.com (0.06), www.literotica.com (0.06), www.nbcnews.com (0.06) |
| FineWeb-Edu | en.wikipedia.org (0.85), en.wikisource.org (0.12), www.reference.com (0.10), www.mdpi.com (0.09), www.britannica.com (0.09), en.m.wikipedia.org (0.08), phys.org (0.08), www.enotes.com (0.08), www.ncbi.nlm.nih.gov (0.07), www.sacred-texts.com (0.07), www.slideshare.net (0.07), ebooks.adelaide.edu.au (0.07), sacred-texts.com (0.07), www.ukessays.com (0.06), www.wisegeek.com (0.06) |
| PPL Correlation (DD) | www.theguardian.com (1.12), issuu.com (1.11), www.nifty.org (0.90), www.fanfiction.net (0.83), openjurist.org (0.68), www.beeradvocate.com (0.68), www.nytimes.com (0.65), www.literotica.com (0.58), www.washingtonpost.com (0.54), www.slideshare.net (0.54), www.mdpi.com (0.48), archive.org (0.45), www.businessinsider.com (0.43), www.barnesandnoble.com (0.43), www.forbes.com (0.43) |
| PPL Correlation (DP) | (www.fanfiction.net, 0.36), (www.opentable.com, 0.32), (www.ncbi.nlm.nih.gov, 0.30), (www.alldiscountbooks.net, 0.29), (www.hindawi.com, 0.26), (pubmedcentralcanada.ca, 0.26), (www.rockpapershotgun.com, 0.23), (www.literotica.com, 0.22), (link.springer.com, 0.20), (politicalticker.blogs.cnn.com, 0.20), (sixstat.com, 0.19), (openjurist.org, 0.19), (www.homeaway.com, 0.17), (www.nifty.org, 0.17), (clinicaltrials.gov, 0.16) |
| DCLM | www.physicsforums.com (0.59), math.stackexchange.com (0.56), www.reddit.com (0.53), slashdot.org (0.47), stackoverflow.com (0.46), physics.stackexchange.com (0.46), en.wikipedia.org (0.44), tvtropes.org (0.35), news.ycombinator.com (0.31), quizlet.com (0.27), everything2.com (0.27), programmers.stackexchange.com (0.24), www.scribd.com (0.23), www.reference.com (0.23), www.freezingblue.com (0.23) |
| PRESELECT | en.wikipedia.org (3.12), www.fanfiction.net (0.74), www.nifty.org (0.69), www.literotica.com (0.45), stackoverflow.com (0.38), en.m.wikipedia.org (0.35), tvtropes.org (0.27), www.sexstories.com (0.25), slashdot.org (0.21), en.wikisource.org (0.21), archiveofourown.org (0.20), sacred-texts.com (0.20), www.sacred-texts.com (0.19), chowhound.chow.com (0.17), lists.w3.org (0.16) |

ence scores include *(Why, 672)*, *(Answer:, 620)*, *(assistant., 567)*, *(Question:, 448)*, *(Describe, 368)...* It is clearly observed that most positive uni-gram features have supervised fine-tuning data patterns and if the data in the pretraining corpus has such uni-gram, they have a high probability to be selected. While for PRESELECT, the uni-gram features with high influence scores include *(ˆ, 338)*, *(MIT, 241)*, *(Math, 194)*, *(answer, 188)*, which widely covers many features appeared in code, education, and question answering domains. For example, "ˆ" exists in many code snippets, "MIT" and "Math" exist in many education-related examples. These representative uni-gram features does not show in domain-level correlation estimation method, Perplexity Correlation (DP), where various diverse meaningless features are learned. We attribute this to the noise in the domain-level correlation calculation, where inadvertently include low-quality data from the high-correlation domains while overlooking high-quality data from unselected domains. The full learned positive and negative uni-grams alongside their influence score can be found in appendix A.3.2 which provides more insights into what word patterns are preferred by different methods.

### 4.3. Distribution of Selected Data

After showing what kind of documents PRESELECT tends to look for in the corpora, we would like to know (1) what data PRESELECT actually select? and (2) How PRESELECT selected data different from the data selected by other base-

lines? We begin by analyzing the data distribution according to the source domain of selected data. As shown in Table 4 , we list top-15 source domains with their percentage density in terms of character number in descending order. We list more domains in Appendix A.5.

For domain distribution of the original corpus where we just randomly sampled a subset, *wikipedia.org* takes the highest percentage and followed by general utility domains such as news (*theguardian*, *nytimes*, *washingtonpost*...) and literature/digital libraries (*fanfiction*, *ntify*, *literotica*...), which spreads quite evenly.

FineWeb-Edu aims to select education-related contents which prioritize many education-related domains such as *wikisource.org*, *britannica.com*, *phys.org*, *ncbi.nlm.nih.gov*, while down-sampled those general utility domains. This makes FineWeb-Edu achieve great performance on examination-related benchmarks such as ARC, hurting the performance on understanding such as LAMBADA. DCLM aims to select SFT-like data and is clearly reflected in the selected data, where a large number of question-answer related domains are up-sampled such as *physicsforums.com*, *stackexchange.com*, *reddit.com*, *stackoverflow.com*. These SFT-like data brings DCLM significant advantages over random-sampled baseline on many downstream tasks. Perplexity correlation (DD) selects domains based on domain-level correlation, from the highest correlation domains until filling up token budgets. This means it will discard all the

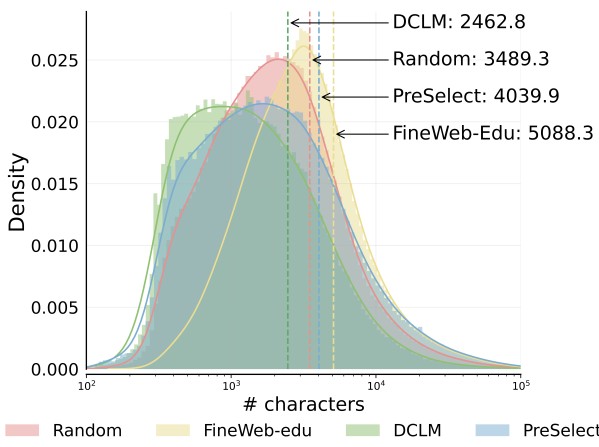

Figure 3: Length distribution of different data selection baselines on RefinedWeb measured by the number of characters. The length annotations are averaged characters.

remaining domains with relatively medium/low correlations once sufficient training tokens are obtained (e.g. only 1500 domains are selected to fill up the required training tokens). As shown in Table 4 row 4, many high-quality domains such as *wikipedia.org* may be dropped according to medium correlation and automatically upsampled each single selected domain, such as *theguardian.com* weights from 0.12% in original data distribution to 1.12% in perplexity correlation (DD). This significantly hurt the data diversity and quality, even leading to a performance drop on HellaSwag, Arc-Challenge compared to random selection, as shown in Appendix Table 12.

Perplexity correlation (DP) utilizes domain-level correlation estimation and train a fasttext based on domain-level positive/negative samples to perform page-level selection. This mitigate the problems in Perplexity correlation (DD) to some extent, but it suffers from the noise inside each domains as we discussed in §4.2. This also reflected in final selected data where the learned features fail to upsample high-quality domains as shown in Table 4.

Different from them, our method tends to select data that correlates with the downstream abilities while broadly containing previously mentioned domains such as *wikipedia.com*, *fanfiction.net*, *stackoverflow.com*, and *wikisource.org*. In addition, our method operates at the document level, which allows for fine-grained filtering of content within each domain and may achieve higher diversity and better quality.

### 4.4. Length Distribution of Selected Data

The data preference difference is also reflected in the length distribution of different methods as shown in Figure 3. The random-sampled data, marked in red, shows a reference distribution of the whole corpus with an average character number of around 3500. While both DCLM and FineWeb-Edu show a significant skew toward short data and long data, with an average length of 2500 and 5100 characters respectively, PRESELECT selects a broader range of data with an average length of 4000, mitigating the risk of bias towards specific document lengths.

## 5. Related Work

**Rule-based Data Selection** Rule-based methods typically employ human-crafted heuristic filters to select data. For example, the C4 pipeline (Raffel et al., 2020) and Gopher rules (Rae et al., 2021) filtered documents by their document length, mean word length, and URL domains. More recently, RefinedWeb (Penedo et al., 2024d) and FineWeb (Penedo et al., 2024a) brought more heuristic quality filters such as character repetition, the fraction of lines ending with punctuation, etc. While straightforward and computationally efficient, these approaches often fail to generalize across diverse domains and contain strong human bias.

**Model-based Data Selection** In contrast to rule-based approaches, model-based data selection leverages trained models to score the data. For instance, CC Net retained the text that is assigned a low perplexity by a language model trained on Wikipedia (Wenzek et al., 2020), Marion et al. (2023) and Ankner et al. (2024) also use perplexity as a practical and effective metric for data pruning, FineWeb-Edu trained a BERT classifier to prioritize educational contents (Penedo et al., 2024b), DsDm and MATES employed data influence models to assess the impact of specific data points on pretraining (Engstrom et al., 2024; Yu et al., 2024). And recently, DCLM (Li et al., 2024a) also uses a fastText-based scorer which selects referring to supervised-finetuning data. These methods offer stronger performance than rule-based methods, however, they either require heavy computation or still contain strong human heuristics. Different from them, PRESELECT is more principled and lightweight, thus bypassing strong human heuristics and offering adaptability.

## 6. Conclusion

In this work, we introduce PRESELECT, a novel predictive data selection tailored for language model pretraining that leverages the predictive strength of the data (i.e. how effectively the normalized losses on the data represent the model's capabilities) to identify high-quality and low-quality data. By operating at the document level, PRESELECT significantly improves granularity and adaptability in data selection. Experimental results across multiple model scales validate the effectiveness of PRESELECT, demonstrating substantial improvements over random selection and recent pretraining data selection baselines in both downstream task performance and computational efficiency.

## Acknowledgments

This project is partially supported by NSFC Grant 62306177.

## Impact Statement

This paper presents work whose goal is to advance the field of Machine Learning. There are many potential societal consequences of our work, none of which we feel must be specifically highlighted here.

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

# A. Filtering Details

This section serves as an extension of § 2, which introduce the detail implementation of our compression-based data selection and fastText classifier training, and pretriaining data filtering.

## A.1. Identified Positive/Negative Data

We first sampled a small subset of data which also from pretraining corpus data distribution but is not from the data pool for later model training. We counted the 3,000 most frequent domains in the pretraining corpus (e.g. *wikipedia.org*) and randomly sampled 300 documents for each domain which result in 900,000 documents in total. We then choose a series open-sourced models to compute compression efficiency. Here we choose Llama-1-7B, Llama-1-13B, Llama-1-30B, Llama-1-65B, Llama-2-7B, Llama-2-13B which cover a wide range in terms of model size. From Huang et al. (2024), we already know that the following order holds in terms of averaged downstream performance:

$$\text{Llama-1-65B} > \text{Llama-1-30B} > \text{Llama-2-13B} > \text{Llama-1-13B} > \text{Llama-2-7B} > \text{Llama-1-7B}$$

So next step, **for each data**, we calculate the compression efficiency **for each model**. This means each data document has corresponding 6 compression efficiency. We than score each document according to the matching between compression efficiency and downstream task performance by Eq. 1. The score distribution is shown in Figure 4 below.

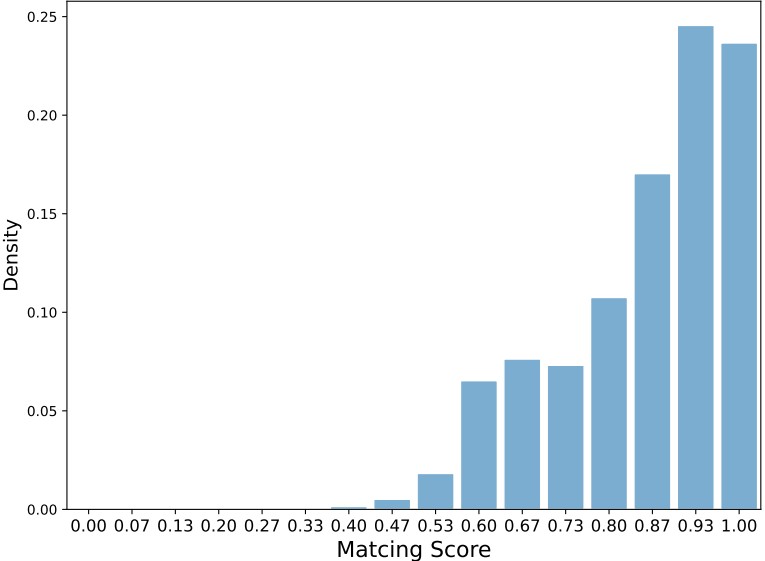

Figure 4: The matching score distribution on RefinedWeb subset where reflect the alignment between compression efficiency and averaged downstream performance. The higher the better.

We then select examples with score = 1 as the positive(good) data and select negative(bad) data gradually from score = 0 as dicussed in § 2.3. We have already give discussion about what kind of data are these positive and negative data in § 4.1, here we list more domains (e.g. top-30 domains) these positive/negative documents from in Figure 5 and Figure 6 respectively.

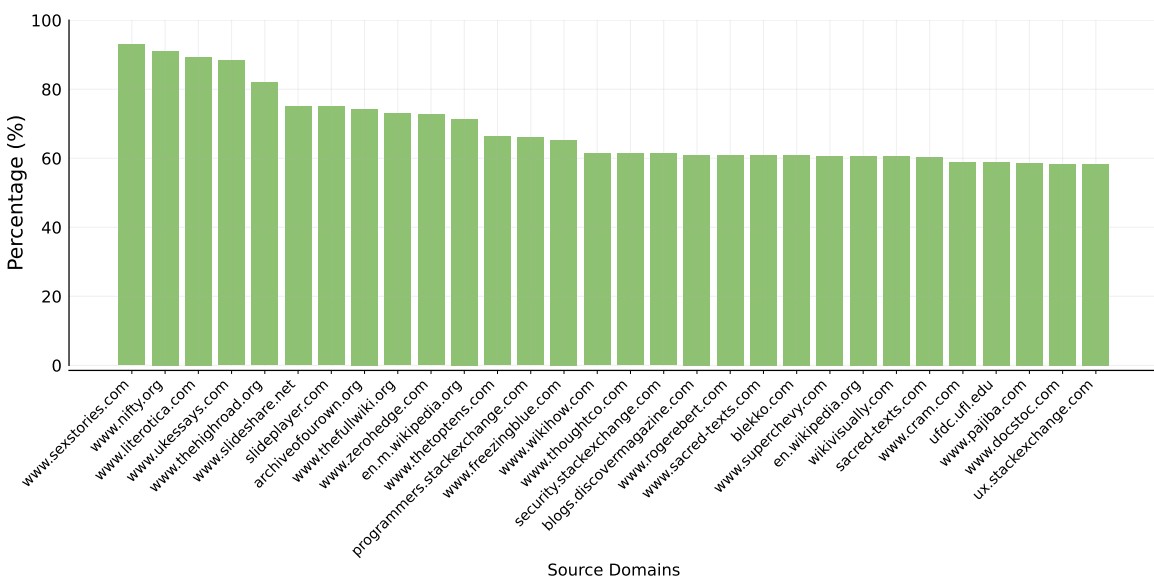

Figure 5: The Top-30 domains where the identified positive documents from, measured by percentage of positive documents inside that domain.

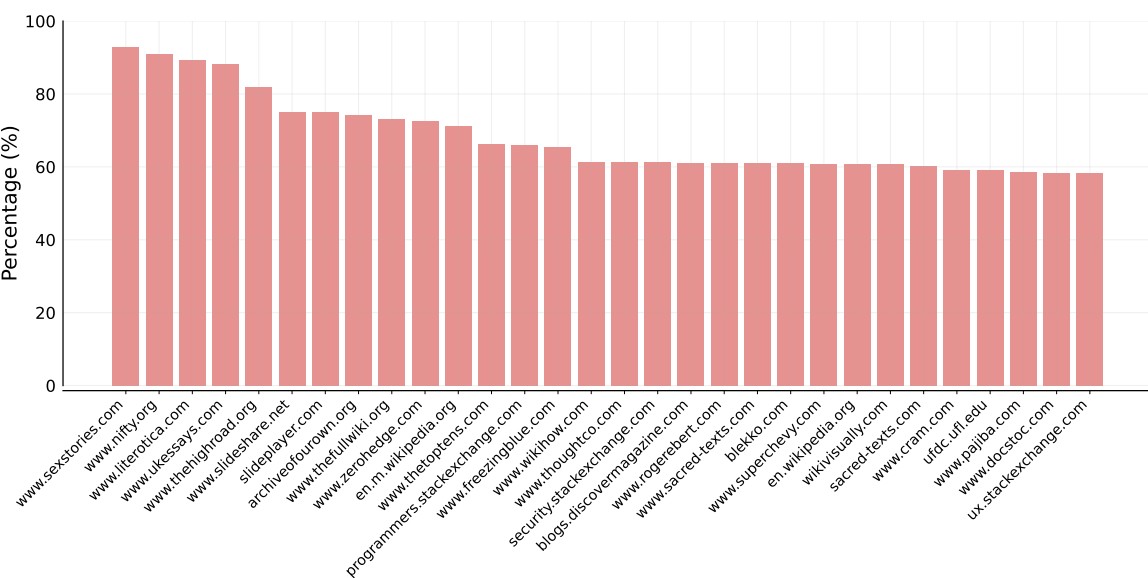

Figure 6: The Top-30 domains where the identified negative documents from, measured by percentage of negative documents inside that domain.

## A.2. FastText Training Details

We use the official fastText python library[3] for training the classifier, some important hyper-parameters are listed in Table 5 below. Basically we follow the default setting of fastText training, however, since both our positive examples and negative examples are from pretraining data distribution which tend to be diverse, so we increase the epoch number to 5 for better trianing convergence.

Table 5: The training hyper-parameters of our fastText classifier.

| | |
|---|---|
| lr | 0.1 |
| dim | 100 |
| epoch | 5 |
| minn | 0 |
| maxn | 0 |
| wordNgrams | 2 |

Specifically we found that fastText will internally add <\s> as an end of sentence token at the end of each documents, which will be learned as a uni-gram feature as well. However, this token/uni-gram feature generally does not exist in the data we want to score (e.g. the pretraining corpus). Given every documents will be affected by this uni-gram feature with weight 1, we found this would potentially bring length bias towards data selection. If <\s> was learned as a negative feature, than short data will be influenced more according the mechanism of fastText scoring calculation, which lead to a general low score of short data. Similarly, if <\s> was learned as a positive feature, short data will also be influenced more, which lead to a high score of short data. To bypass such bias towards length, we manually set 0 weight to this uni-gram feature.

## A.3. Learned FastText Feature

### A.3.1. PRELIMINARY - FASTTEXT FEATURE VISUALIZATION

According to the fastText classifier model architecture (Joulin et al., 2017), there are basically two components - Input Matrix and Output Matrix, where input matrix maps all uni-gram feature to hidden dim (e.g. 100) and output matrix maps hidden dim to final classification dimension, in our case, is 2. Finally, a softmax layer is applied to obtain the classification score for positive label and negative label.

Thus we can calculate the importance of each feature (uni-gram). Specifically, we define the input with $N$ n-grams features $x_1, x_2, ... x_N$, which represent the occurrence of each individual n-gram feature. We then define input matrix to be $A \in \mathbb{R}^{100 \times N}$ for simplicity where 100 is the hidden dimension size and output matrix to be $B \in \mathbb{R}^{2 \times 100}$. Thus the final layer is

$$BA \begin{bmatrix} x_1 \\ x_2 \\ ... \\ x_N \end{bmatrix} = \begin{bmatrix} y_1 \\ y_2 \end{bmatrix} \tag{2}$$

where $y_1$ is the value before softmax for positive label and $y_2$ is the value before softmax for negative label. Thus, for each individual feature $x_i$, we can calculate its contribution to $y_1$ and $y_2$ given our trained $A$ and $B$. Specifically, for each feature $x_i$ we can calculate $BA \begin{bmatrix} 0 \\ ... \\ x_i \\ ... \\ 0 \end{bmatrix} = \begin{bmatrix} y_{i1} \\ y_{i2} \end{bmatrix}$ where $x_i = 1$. And $(y_{i1} - y_{i2})$ is used as the feature importance where a positive value represents a positive feature.

Another problem for such importance visualization is that we train the fastText classifier with *Ngram* equals 2 where bi-gram features are hashed into buckets which prevent us from tracking the importance of specific uni-gram or bi-gram features.

---

[3]https://github.com/facebookresearch/fastText

However, we found for a fastText classifier trained with uni-gram + bi-gram features, if we only give its uni-gram and make the final score prediction, that does not have a huge difference to using all features (with < 0.02 score difference out of 1.0). Then we explore the feature importance for uni-gram only.

### A.3.2. FASTTEXT FEATURE INFLUENCE SCORE

We briefly discussed the learned features with their influence scores in § 4.2, and we list the top-50 positive/negative features alongside the influence scores in Table 6 below.

Table 6: The Top-50 fastText learned fearures with corresponding influence scores of different methods. Positive and Negative are defined according to the magnitude of the influence score.

| Method | Top-50 fastText Features with Influence Scores |
|---|---|
| DCLM (pos) | (""', 968.194), (Why, 671.558), (Answer:, 620.404), (Additionally,, 599.6), (assistant., 567.250), (Edit:, 565.787), (Generate, 537.11), (answer., 502.883), (Given, 474.243), (A:, 457.440), (Question:, 447.825), (Write, 440.69), (task., 423.878), (ELI5, 419.001), (AI, 403.29), (question:, 394.996), (\\\\, 378.738), (why, 378.032), (answering, 370.485), (Describe, 368.243), (explanation., 349.408), (basically, 347.061), (sentence, 338.57), (&, 331.964), (answer, 328.221), (assistant,, 324.112), (How, 322.649), ([deleted], 320.465), ([Your, 317.477), (Overall,, 316.189), (Reddit, 315.919), (EDIT:, 314.785), (explain, 307.202), (Provide, 292.085), (step-by-step, 290.89), (old., 283.723), (Rewrite, 280.710), (user], 280.342), ([deleted, 279.535), (Title:, 279.145), (is:, 278.694), (can., 278.337), (Identify, 270.603), (Explain, 268.238), (Imagine, 259.265), (Summarize, 254.946), (Create, 254.661), ([removed], 254.198), (Roleplay, 253.866), (Whats, 250.611) |
| DCLM (neg) | (–, -479.818), (Comment, -466.0545), (Posted, -450.19833), (Post, -442.57623), ([. . . ], -386.1609), (Comments, -377.41232), (—, -344.9523), (About, -340.93152), (. . ., -321.9254), ("The, -321.84586), (Re:, -310.0053), (More, -297.77707), (», -294.49173), (See, -289.7501), (2012, -278.6206), (View, -278.62042), (PM, -274.32016), (©, -272.98975), (2017, -269.48788), (Read, -267.5124), (Leave, -265.90674), (Related, -260.5703), (2013, -260.09442), (Love, -254.712), (Free, -247.34369), (Home, -247.21219), (2014, -244.1095), (Q:, -240.64737), (Click, -239.72502), (Questions, -238.72769), (blog, -232.80008), (&, -231.04167), (2010, -225.71875), (Be, -225.16672), (Buy, -222.93646), (Originally, -219.75381), (..., -218.0946), (Last, -217.36478), (Search, -216.1242), (2011, -214.54568), (·, -214.43338), (by:, -213.91516), (to:, -213.87567), (Don't, -212.08264), (Get, -210.06216), (2019, -207.97916), (Continue, -203.56561), (Although, -203.0502), (Date:, -201.52185), (2009, -199.80365) |
| PRESELECT (pos) | (ˆ, 337.7), (Likes, 241.387), (shareimprove, 241.28), (MIT, 240.614), (–\xa0, 240.006), (Retrieved, 232.997), (Received, 214.872), (Photo:, 205.808), (Azure, 200.453), (Tolkien, 195.01), (Math, 193.741), (answer, 187.706), (Contributed, 185.518), (Cite, 184.729), (MATLAB, 176.991), (Document, 172.13), (Youre, 168.549), (API, 168.372), (Rhapsody, 166.203), (Name, 165.407), (Novell, 165.201), ((for, 162.172), (Dismiss, 159.38), (Geek, 156.977), (Examples, 153.852), (Expand, 153.352), (add-ons, 153.074), ((and, 151.961), (Techdirt, 151.241), (Perl, 148.295), (12, 147.958), (Youll, 146.352), (Random, 144.253), (macrumors, 142.24), (p., 141.384), (Fool, 140.61), (secs, 140.129), (1., 138.052), (bass, 136.025), (Users, 132.508), (Boards, 132.233), (IBM, 131.088), (Journey, 130.867), (Step, 130.648), (BrainyQuote, 130.109), (of\xa0the, 127.833), (Preview, 127.324), (Comentários, 126.514), (contributors, 126.057), (Early, 124.897) |
| PRESELECT (neg) | (Register, -375.421), (Details, -335.247), (Comments, -320.821), (Updated:, -310.755), (COVID-19, -303.878), (tho, -281.664), (Print, -271.024), (Profile, -262.205), (Leave, -258.85), (Published:, -235.235), (Product, -228.646), (coronavirus, -227.227), (2020, -223.704), (By:, -222.939), (Shipping, -219.518), (Share, -219.401), (Login, -219.266), (Deals, -211.159), (Add, -207.725), (item, -201.697), (Reviews, -201.59), (Sponsored, -200.319), (Current, -196.23), (», -195.792), (Follow, -194.77), (2021, -192.621), (©, -192.304), (Join, -189.937), ("", -189.647), (Post, -186.328), (View, -183.529), (Item, -182.289), (Privacy, -176.569), ("We, -176.348), (Search, -175.912), (Publication, -175.804), (Author:, -173.699), (Subscribe, -170.517), (Contact, -165.304), (ISBN, -163.925), (About, -163.91), (account?, -163.671), (reviews, -162.936), (Posted, -162.205), (Customer, -160.427), (Level, -155.532), (Paperback, -155.16), (Author, -153.952), (Submit, -152.988), (Ratings, -151.985) |

### A.4. Extend to Pretraining Corpora

We use DataTrove (Penedo et al., 2024c) which is a library for processing data at very large scale. The final filtering process on pretraining corpora does not need any GPU resrouces and the filtering process is done on the machine with 4 AMD EPYC 9654 96-Core Processor.

### A.5. Selected Pretraining Data Details

As an extension of Table 4, we plot the selected data distribution in Figure 7 where we use the same y-axis scale for all four sub-plots which means the magnitudes of the bars are directly comparable. And we further include the selected pretrianing data distribution of perplexity correlation (Thrush et al., 2024) which uses domain-level correlation and then selects domains in Figure 8 below.

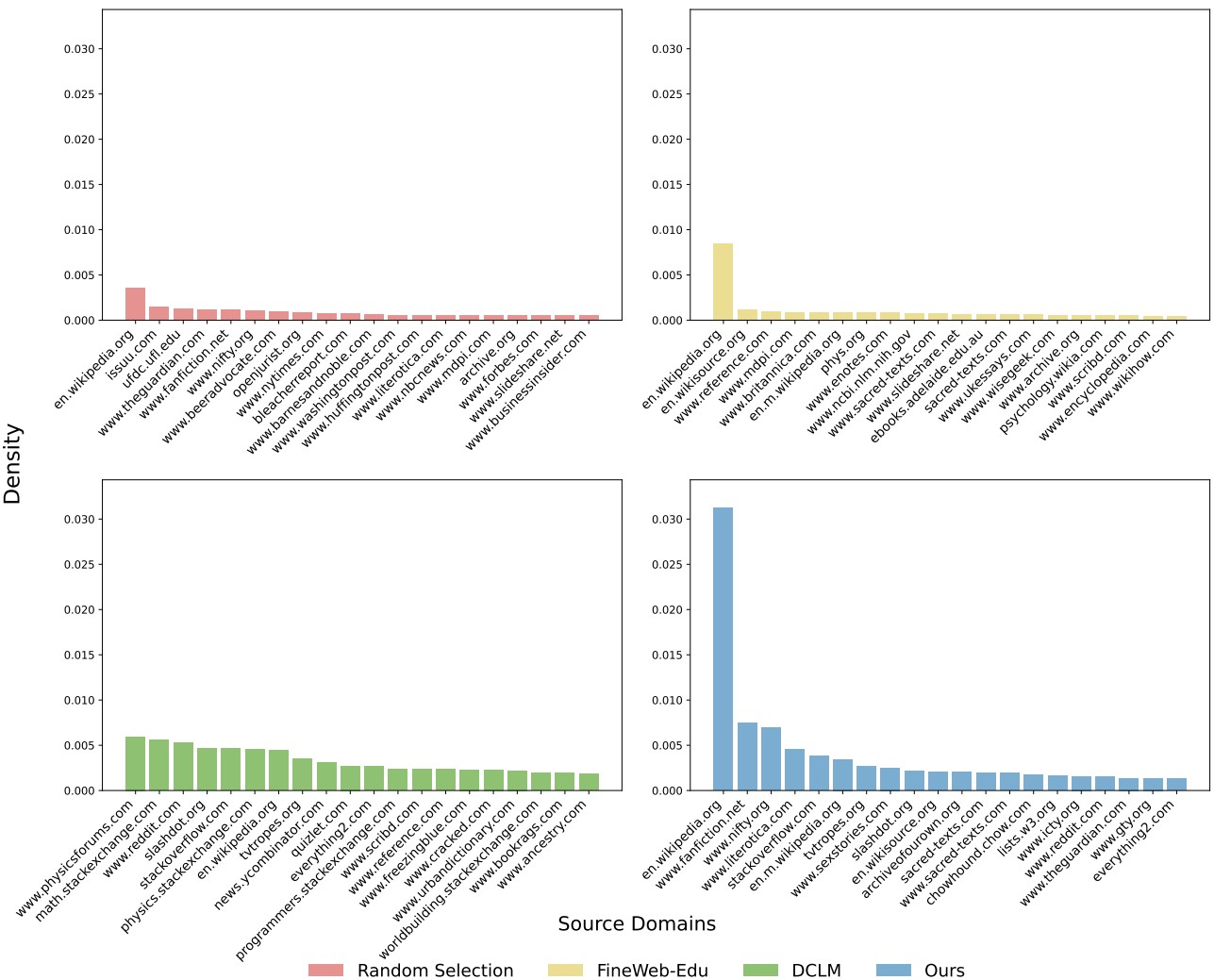

Figure 7: The selected data distribution (RefinedWeb) over source domains of different data selection baselines in descending order. The top-20 domains are listed for comparison and the density are counted based on character percentage. The y-axis scale are same across four subplots.

### A.6. Filtering Computation Overhead

Similar to many model-based methods such as MATES, DsDm (Yu et al., 2024; Engstrom et al., 2024), PRESELECT involves pre-computing the compression efficiency on a small set of data. As we show in § 2.3, we compute compression on around 0.6B tokens by Llama models ranges from 7B to 65B only once for all experiments. Accoding to Kaplan et al. (2020), the inference FLOPs is $C_{\text{infer}} \approx 2 \cdot N(D) = 1.8 \times 10^{20}$ FLOPs, equivalent to 25 H100 hours, which is small, acceptable even negligible when training a large model.

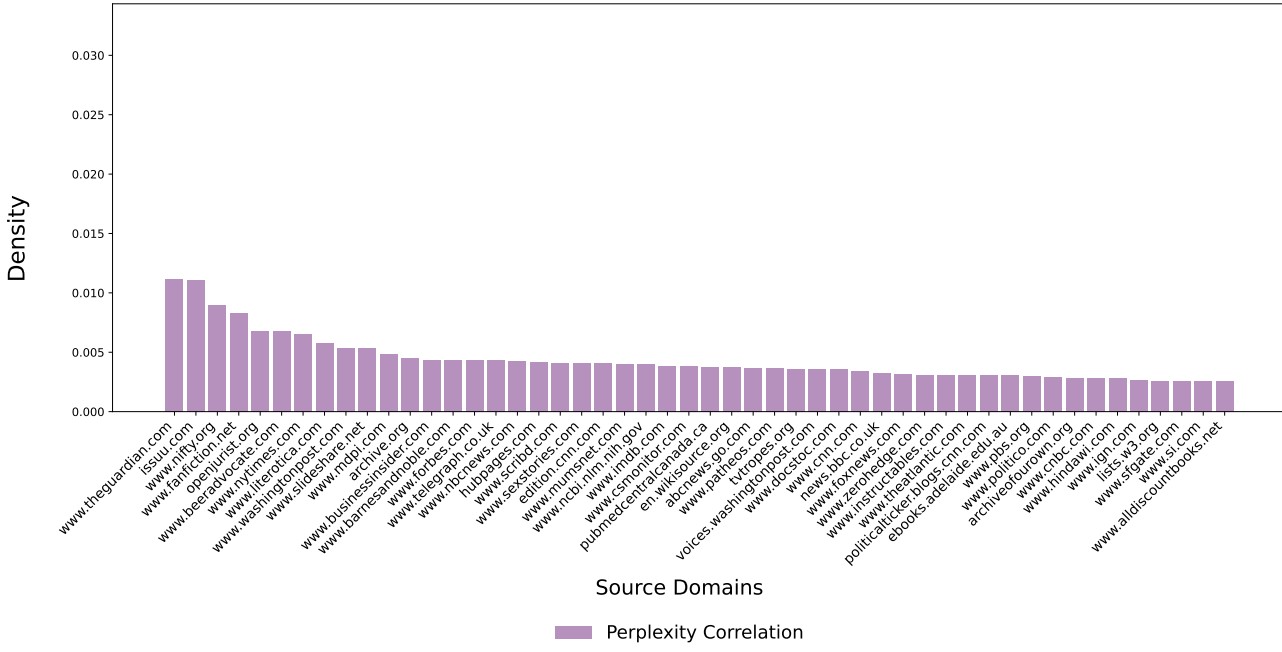

Figure 8: The selected data distribution(RefinedWeb) over source domains of Perplexity Correlation in descending order. The Top-50 domains are listed for comparison and the density are counted based on character percentage. The y-asis scale are same as Figure 7.

## A.7. Additional Analysis on Filtering

### A.7.1. COMPARISON WITH SCALINGFILTER

ScalingFilter (Li et al., 2024b) uses the perplexity difference between a large model and a small model as a metric to select data, which can be viewed as a method only contrasts losses between two model scales. To show these intermediate level models actually help, we compare with ScalingFilter under our setting, using llama-65B and llama-7B to select data under ScalingFilter while removing the mid-size models to compute the Quality factor. Results of training 1B model on 30B tokens with our data pool (Table 1 setting) are shown in Table 7 below.

Table 7: The averaged performance comparison between ScalingFilter and PRESELECT.

| Method | Average Benchmark Performance |
|---|---|
| Random | 37.2 |
| ScalingFilter | 37.6 |
| PRESELECT | **40.3** |

From Table 7, we see that after removing the 4 mid-size models, the results drop by 2.7 points on average. Take a further step, we observe that data with a significant perplexity difference is skewed toward shorter texts, with an average character length of 2300, compared to the original corpus distribution's average of 3500 characters. We further see that the top domains where these data with significant perplexity difference from contains (*regator.com*, *msbusiness.com*, *www.yourobserver.com*, *www.qacollections.com*, *www.stylebistro.com*, *libn.com*, *www.businessmanagementdaily.com*) where none of typical high-quality domains exist. These evidence suggests that using the perplexity difference between a large model and a small model tends to select easy and short data.

In addition, a large loss difference between the largest model and the smallest model does not necessarily imply a rank preservation among a series of models. On the other hand, a rank matching between losses and downstream performance of a series of models does not necessarily imply a large losses difference between the largest model and the smallest model. For example, we calculate the spearman correlation between the ScalingFilter perplexity ratios and our PRESELECT predictive strength, which is 0.0533. And they also have a Pearson correlation of -0.079 which indicate a low correlation between these two metrics (measured based on the actual predictive strength score where we calculate based on the sampled subset), indicating the effectiveness of intermediate level models.

### A.7.2. TARGETING SPECIFIC DOWNSTREAM TASK

Since our setting is pre-training, we didn't intend to choose a specific task as the target. However, in our initial experiments, we did explore such differences. For example, when choosing HellaSwag to represent intelligence that led to a different model ranking from using the average. This results in a performance improvement on knowledge-intensive tasks, indicated by 5% lower losses on wiki-related raw texts while having much higher (worse) losses on other domains such as 8% on math and 16% on code, at a 400M model and 8B token training scale. Afterwards, we consistently choose the average performance to represent intelligence under pre-training setting, but We think this is evidence that our method actually predicts "downstream abilities" beyond only pre-training scale.

### A.7.3. MITIGATING SENSITIVE CONTENTS

In our observation, literature and knowledge related contents are more reflective of downstream tasks (generally have a better rank alignment) which includes essays and some adult contents. We believe filtering adult content is an orthogonal direction that could be done separately, which is not the focus of this work. For example, we think a rule-based pre-processing step to filter adult contents out could greatly mitigate this issue.

# B. Pretraining Details

For pretraining, mainly we have two settings, one for our main experiments (Refinedweb (Penedo et al., 2024d)) and another one follows MATES (Yu et al., 2024) which use Pythia (Biderman et al., 2023) architecture with C4 (Raffel et al., 2020) as the training corpora. And we will separately discuss their details in the following subsections.

## B.1. Pretraining Infra

**Framework**   For our main experiments on RefinedWeb, we follow MAP-NEO (Zhang et al., 2024a) and adapted a Megatron-based (Shoeybi et al., 2020) training framework which allows us to efficient training models with different sizes under single-node or multi-node. Since the largest model size is 3B, so we do not have to use any tensor parallelism or pipeline parallelism. While for the experiments under C4, following MATES, we use litgpt (AI, 2023), which is the training framework used by TinyLlama (Zhang et al., 2024b).

**Resource**   For our pre-training, we mainly use 8 H800 (1 node) for training 1B models. While for some relatively large experiments such as 3B models, we use 4 nodes ×8 H800 distributed training.

## B.2. Pretraining Corpus

Given the different size of corpus and different source of corpus across our whole experiments, we list our pretriaining corpus setting for each part of experiments alongside their selection ratio, number of tokens etc. in Table 8.

Table 8: Pretraining corpus details for different experiments. *: which subset of the whole corpus is used in code.

| Table | Corpus | Subset From* | Method | Model Size | Pool Size | Selection Ratio | Trained Tokens |
|---|---|---|---|---|---|---|---|
| Table 1 | RefinedWeb | $k \in [0, 10]$ | Random Selection
DCLM
PRESELECT | 400M | 80B | N/A
10%
10% | 8B |
| | | $k \in [0, 40]$ | Random Selection
Random Selection
Perplexity Filter
FineWeb-Edu
DCLM
Perplexity Correlation(DD)
Perplexity Correlation(DP)
PRESELECT | 1B | 300B | N/A
N/A
10%
10%
10%
10%(domain-wise)
10%
10% | 30B
300B
30B
30B
30B
30B
30B
30B |
| | | | DCLM
PRESELECT | | | 30% | 90B |
| | | | DCLM
PRESELECT | | | 50% | 150B |
| | | $k \in [0, 130]$ | Random Selection
DCLM
PRESELECT | 3B | 1T | N/A
10%
10% | 100B |
| Table 2 | C4 | N/A | PRESELECT | 410M
1B | 200B | top-25B | 25B |

For RefinedWeb, we use a version[4] that after heuristic filtering and deduplication following DCLM and randomly sampled a subset of needed number of tokens. We evenly sampled from each global index and local index. For C4, we use the whole dataset, which is about 198B tokens.

---

[4]https://data.commoncrawl.org/contrib/datacomp/DCLM-refinedweb/index.html

## B.3. Model Architecture

Since we train the models from scratch, we list our model architectures in Table 9. For fair comparison, different data selection baselines we compared at the same scale (e.g. 1B) use the same model architecture.

Table 9: The pretraining model architecture details. For more configuration, please refer to our code repo.

| Model Size | # Heads | # Layers | Context Length | Vocabulary Size | Hidden Size | FFN Hidden Size |
|---|---|---|---|---|---|---|
| NEO Architecture | | | | | | |
| 400M | 8 | 12 | 4,096 | 64,005 | 1,024 | 8,192 |
| 1B | 8 | 12 | 4,096 | 64,005 | 1,536 | 12,288 |
| 3B | 8 | 24 | 4,096 | 64,005 | 2,048 | 16,384 |
| Pythia Architecture | | | | | | |
| 410M | 16 | 24 | 1,024 | 50,304 | 1024 | 4096 |
| 1B | 8 | 16 | 1024 | 50,304 | 2048 | 8192 |

## B.4. Pretraining Hyperparameters

For pretraining hyperparameters, consistent with many open-sourced models, we consistently use a batch size 1,048,576 tokens, which is 4096 context length × 256 global batch size. Also widely used AdamW optimizer and cosine decay learning rate schedular are used. For Pythia models, we try the best to keep the same training setting with MATES (Yu et al., 2024).The detailed training hyperprameters are listed in Table 10 below.

Table 10: The detailed pretraining hyperparameters for all experiments. For more configuration, please refer to our code repo.

| Model Size | Batch Size(Token) | Global Batch Size | Learning Rate | Schedular | LR Warmup | Optimizer | TP/PP |
|---|---|---|---|---|---|---|---|
| NEO Architecture | | | | | | | |
| 400M | 1M | 256 | 2e-4 → 2e-5 | cosine | 0.5% | AdamW | 1/1 |
| 1B | 1M | 256 | 2e-4 → 2e-5 | cosine | 0.5% | AdamW | 1/1 |
| 3B | 1M | 256 | 2e-4 → 2e-5 | cosine | 0.5% | AdamW | 1/1 |
| Pythia Architecture | | | | | | | |
| 410M | 500K | 1,024 | 1e-3 → 6.25e-5 | WSD | 2,000 | AdamW | 1/1 |
| 1B | 500K | 1,024 | 1e-3 → 6.25e-5 | WSD | 2,000 | AdamW | 1/1 |

# C. Evaluation Details

## C.1. Evaluation Framework

For our evaluation framework, we mainly adapt two widely used framework Opencompass (Contributors, 2023) and LM-Evaluation-Harness (Gao et al., 2024) to cover a broader range of tasks as either one may lack some tasks in specific field. For Math and Code domain, we adapted the code from Huang et al. (2024) and calculating the bpc based on provided math and code raw texts (e.g. Math Arxiv paper and GitHub code). Considering some bugs of these framework to run specific task, we list our used evaluation details, including evaluation method, metric, framework etc. for each task in Table below. We also categorize each task into an ability domain according to opencompass which provides a better visualization.

Table 11: The full details of evaluation framework, mode and metrics for each task. *: we adapt the code from Huang et al. (2024) for BPC computation.

| Domain | Task | Evaluation Mode | Metric | Framework |
|--------|------|-----------------|--------|-----------|
| Examination | Arc-Easy | Perplexity | Accuracy | Opencompass |
| | Arc-Challenge | Perplexity | Accuracy | Opencompass |
| | MMLU | Perplexity | Accuracy | Opencompass |
| Understanding | LAMBADA | Generation | Accuracy | Opencompass |
| | OpenBookQA | Perplexity | Accuracy | Opencompass |
| | RACE-Middle | Perplexity | Accuracy | Opencompass |
| | RACE-High | Perplexity | Accuracy | Opencompass |
| | MultiRC | Perplexity | Accuracy | Opencompass |
| Reasoning | HellaSwag | Perplexity | Accuracy | Opencompass |
| | PIQA | Perplexity | Accuracy | Opencompass |
| | SIQA | Perplexity | Accuracy | Opencompass |
| | SciQ | Perplexity | Accuracy | LM-Evaluation-Harness |
| | RTE | Perplexity | Accuracy | Opencompass |
| | BBH | Generation | Exact-Match | LM-Evaluation-Harness |
| Language | WinoGrande | ll | Accuracy | Opencompass |
| Math | Math ($\downarrow$) | loss | BPC | * |
| Code | Code ($\downarrow$) | loss | BPC | * |

## C.2. Full Evaluation Results

Due the page limitation of main body, we are unable to include all 17 task performance in Table 1 in § 3.4. Here we list the full evaluation results of our method and various data selection baselines on 17 tasks, grouped by model size. The full evaluation results of 1B models and 3B models are presented in Table 12 and Table 13 respectively.

Table 12: Full evaluation results of various data selection baselines and our method on 17 tasks at 1B scale × 30B token.

| Task | Random Selection | Perplexity Filter | FineWeb-Edu | Perplexity Correlation(DD) | DCLM | PRESELECT |
|---|---|---|---|---|---|---|
| Arc-Easy | 39.2 | 42.5 | **48.3** | 39.7 | 45.2 | 48.0 |
| Arc-Challenge | 24.4 | 24.6 | 26.1 | 23.7 | 24.8 | **26.8** |
| MMLU | 26.0 | 25.8 | 26.0 | 26.1 | **26.3** | 26.0 |
| LAMBADA | 19.0 | 18.8 | 18.2 | 20.7 | 22.2 | **23.5** |
| OpenBookQA | 30.0 | 30.2 | 30.6 | 29.0 | **31.2** | 31.0 |
| RACE-Middle | 21.5 | 22.5 | 24.8 | 21.9 | 24.6 | **27.9** |
| RACE-High | 22.4 | 22.7 | 23.9 | 23.7 | 24.1 | **27.5** |
| MultiRC | **57.2** | **57.2** | **57.2** | **57.2** | **57.2** | **57.2** |
| HellaSwag | **40.0** | 38.5 | 40.0 | 36.3 | 38.9 | 38.9 |
| PIQA | **69.2** | 67.6 | 67.3 | 67.6 | 67.6 | 67.7 |
| SIQA | 32.1 | 32.5 | 33.6 | 33.5 | 33.9 | **34.9** |
| SciQ | 64.8 | 67.5 | 69 | 63.7 | 70.0 | **71.5** |
| RTE | 52.7 | 52.7 | 52.0 | 52.7 | 52.7 | **53.8** |
| BBH | 7.8 | 8.5 | 12.8 | 9.5 | 12.6 | **16.2** |
| WinoGrande | 51.6 | 51.5 | 51.2 | 50.4 | 51.4 | **53.0** |
| **Average** | 37.2 | 37.5 | 38.7 | 37.1 | 38.8 | **40.3** |
| Math (↓) | 1.023 | 0.957 | 0.906 | 0.980 | 0.857 | **0.830** |
| Code (↓) | 0.901 | 0.853 | 0.816 | 0.919 | 0.773 | **0.744** |

Table 13: Full evaluation results of various data selection baselines and our method on 17 tasks at 3B scale × 100B token. **Bold** denotes the best.

| Task | Random Selection | DCLM | PRESELECT |
|---|---|---|---|
| Arc-Easy | 51.2 | 55.7 | **61.2** |
| Arc-Challenge | 29.2 | 31.2 | **31.9** |
| MMLU | 24.8 | 25.3 | **26.2** |
| LAMBADA | 33.2 | 35.1 | **36.1** |
| OpenBookQA | 34.0 | **38.2** | 36.0 |
| RACE-Middle | 22.1 | 25.1 | **26.0** |
| RACE-High | 22.8 | **27.0** | 25.5 |
| MultiRC | **57.2** | **57.2** | **57.2** |
| HellaSwag | 57.5 | 58.4 | **59.2** |
| PIQA | **75.2** | 74.1 | 74.2 |
| SIQA | **35.8** | 33.4 | 33.8 |
| SciQ | 79.5 | 82.5 | **85.6** |
| RTE | 52.4 | **52.7** | 52.4 |
| BBH | 15.3 | 20.5 | **23.3** |
| WinoGrande | 54.3 | **55.8** | 55.0 |
| **Average** | 43.0 | 44.8 | **45.6** |
| Math (↓) | 0.818 | 0.712 | **0.694** |
| Code (↓) | 0.726 | 0.664 | **0.648** |

