# OpenReview forum: "Predictive Data Selection: The Data That Predicts Is the Data That Teaches"
_ICML.cc/2025/Conference — ICML 2025 poster_

### Official Review · Reviewer_xzzW · 2025-03-09

**Overall Recommendation:** 2

**Summary:**

The authors propose a new data selection based on the rankings of perplexity which a range of llama models assign to documents. The method is scaled to large datasets via means of a fasttext filter. The authors claim that this captures which training examples are predictive of broad downstream abilities. The data selected by this method achieves better downstream results than a number of baselines when evaluated in a comparable setting. The authors perform some high-level inspection of domains upweighted and downweighted by the different methods.

**Claims And Evidence:**

While the experimental results look impressive, I have major concerns about the justification and motivation for the method.

**Confounding factors**

The central thesis is that "data on which model losses are predictive of downstream abilities also contribute effectively to learning". However, their method only vaguely involves “prediction of downstream abilities” as they employ a coarse-grained rank-correlational method, where the average benchmark ranking of a small set of models is compared to the models’ perplexity ranking on a specific document.

Their “downstream abilities” model ranking is fixed across all experiments:

Llama-1-65B > Llama-1-30B > Llama-2-13B > Llama-1-13B > Llama-2-7B > Llama-1-7B

which suggests the equally valid hypotheses that either “*documents on which additional model parameters benefits loss are useful for pre-training*” or  “*documents on which additional pre-training compute benefits loss are useful for pre-training.*. The first hypothesis has previously been explored by ScalingFilter [1], which I believe is an important point of comparison for the proposed method. Their method only contrasts losses between two model scales, and you could show that intermediate levels actually help.

To show that the method actually predicts “downstream abilities” beyond only pre-training scale, the authors should choose a setting where different downstream tasks elicit different model rankings. Note that this fine-grained downstream analyses is performed in Perplexity Correlations [2], which the authors discuss as the most similar prior work (although I think that ScalingFilter [1] is actually the most directly related paper).


**Importance of group-level selection**

The authors claim that sample-level selection is an important factor to their success of their method compared to Perplexity Correlations [2], which only performs group-level selection. However, I believe these methods are very different in nature. Therefore, the empirical justification for this claim should be to apply their method and compute perplexity rankings on a group level to select data, rather than comparing to the results of Perplexity Correlations.


[1] Li et al., EMNLP 2024. ScalingFilter: Assessing Data Quality through Inverse Utilization of Scaling Laws

[2] Thrush et al., ICLR 2025. Improving Pretraining Data Using Perplexity Correlations

**Essential References Not Discussed:**

As mentioned above, the paper misses a comparison and discussion of ScalingFilter [1] and Perplexity Pruning [3, 4].

**Experimental Designs Or Analyses:**

The experimental design follows standard practices from prior works. The choice of downstream tasks is also good. However, besides the missing comparisons discussed above,

**Add few-shot performance**

It is not clear whether the main results in Table 1 are from zero-shot or few-shot prompting (whereas Table 2 is explicitly zero-shot). I would suggest adding few-shot results in the appendix at least.

**Low performance**

I'm confused why the general results in Table 1 are so low. For example, a 1B Random selection baseline on 25B tokens on C4 performs in Table 2 outperforms the 30B token DCLM baseline in Table 1 on ARC-easy, ARC-challenge and HellaSwag, despite the known strong performance of DCLM-baseline over C4. The low performance of the 1B model at 300B token is also a bit strange, including the decrease in MMLU score compared to 30B tokens.

**Additional ablations**

The proposed data selection objective is only instantiated with a single fairly arbitrary series of existing models. I would guess that the proposed rank coefficient is quite sensitive to the number and kind of models used. I think it is essential to introduce ablations to understand some of the implicit assumptions on reference models and demonstrate the robustness of the model, i.e. how would the results change when including or excluding large models and when using a different reference model family (e.g. Pythia).
I realize that every type of ablation would be computationally too expensive, but even showing analyses like overlap and differences in selected data would be valuable to the reader.

**Methods And Evaluation Criteria:**

The method makes sense and the evaluation/experimental setting is reasonable for evaluating training data selection. Scaling the document selection to large corpora with fasttext is also understandable.

**Reliance of large pre-trained models**

One potential issue with the experiments is that they rely on much larger models (from 7B-65B) for selecting training data of a small model (1B).
This may limit the utiliy of this method when wanting to scale to even larger models.
Note for example, that recent work in perplexity filtering [3] (another very relevant paper) highlights the use of a smaller model for evaluating perplexity than used for training.
I would suggest repeating the method with smaller models, for example the pythia models of sizes 14M, 31M, 70M, 160M and 410M.


[3] Ankner et al., ICLR 2025. Perplexed by Perplexity: Perplexity-Based Data Pruning With Small Reference Models

**Other Comments Or Suggestions:**

* Equation (1) would be much clearer if you indicated that the score S and the ranks C's are per document!
* Btw Figures 6 and 7 have the same data, despite meaning to show the most negative and positive domains in terms of selected data

**Other Strengths And Weaknesses:**

n/a

**Questions For Authors:**

* I'm curious why you don't address the fact that the method seems to have a strong bias towards selecting ''adult content`` (in Figure 3)

**Relation To Broader Scientific Literature:**

In my view, the key contribution is that the paper provides further empirical evidence that
the loss from pre-trained models can be useful signal for selecting pre-training data,
following from work on using loss from one, two or many models to select data ([1], [2], and [3, 4] respectively). The authors achieve strong results with their method, and beat heuristic quality-based data selection baseline, highlighting the promise of automatic loss-based methods requiring less human intervention. However, there is no discussion or comparison to [1, 3, 4] in the paper.

[1] Li et al., EMNLP 2024. ScalingFilter: Assessing Data Quality through Inverse Utilization of Scaling Laws

[2] Thrush et al., ICLR 2025. Improving Pretraining Data Using Perplexity Correlations

[3] Ankner et al., ICLR 2025. Perplexed by Perplexity: Perplexity-Based Data Pruning With Small Reference Models

[4] Marion et al., 2023. When Less is More: Investigating Data Pruning for Pretraining LLMs at Scale

**Theoretical Claims:**

n/a

---

> ### Author Rebuttal · Authors · 2025-04-01
>
> Dear Reviewer xzzW,
>
> Thank you for your valuable suggestions and insightful comments!
>
> We address your comments one by one as follows:
>
> ---
>
> Q1.[ScalingFilter [1], which I believe is an important point of comparison for the proposed method]
>
> Thanks for pointing out a related work ScalingFilter[1] which uses the perplexity difference between a large model and a small model as a metric to select data. To show that the intermediate levels of models actually help, we compare with ScalingFilter under our setting, using llama-65B and llama-7B to select data under ScalingFilter while removing the mid-size models. Results of training 1B model on 30B tokens with our data pool (Table 1 setting) are below.
>
> |**Method**|**Average Performance**|
> |:-:|:-:|
> |Random |37.2|
> |ScalingFilter| 37.6|
> |PreSelect|**40.3**|
>
>
> We see that after removing the 4 mid-size models, the results drop by 2.7 points on average. We will add these results to the next revision of the paper.
>
> ---
>
> Q2. [choose a setting where different downstream tasks elicit different model rankings]
>
> Since our setting is pre-training, we didn’t intend to choose a specific task as the target which was analyzed by Perplexity Correlation[2].
>
> However, in our initial experiments, we did explore such differences. For example,  we chose a HellaSwag to represent intelligence that led to a different model ranking from using the average. This results in a performance improvement on knowledge-intensive tasks, indicated by 5% lower losses on wiki-related raw texts while having much higher (worse) losses on other domains such as  8% on math and 16% on code, at a 400M model and 8B token training scale. We think this is evidence that our method actually predicts “downstream abilities” beyond only pre-training scale.
>
> ---
>
> Q3. [Importance of group-level selection]
>
> Thank you for your suggestions, we further compare with computing perplexity rankings on a group level to select data. The results and analysis can be seen in Q6,7 of the response to reviewer 95of.
>
> ---
>
> Q4. [Add few-shot performance]
>
> Thank you for your suggestions! We apologize for this confusion. For experiments in Table 1, some of these are zero-shot while some are few-shot. We follow the common practice to use zero-shot for classification tasks (perplexity-based metric), and few-shot for generation tasks. We will indicate this more clearly in the revised version.
>
> ---
>
> Q5. [Low performance in Table 1]
>
> We believe this performance difference stems from differences in the experimental configurations between Tables 1 and 2, including variations in pre-training corpus (DCLM pool from CC vs. C4, quality differs in each subset), training framework, different evaluation framework (prompt), slightly different model size etc. Thus these two tables are not that comparable. Regarding the “strange” MMLU results from 1B models in Table 1, those numbers are all around random guess (25%) and do not reflect much, because MMLU is too difficult for small models to learn quickly. For example, in DCLM’s paper, even a 7B model trained at chinchilla optimal scale still shows a nearly random accuracy on MMLU.
>
> ---
>
> Q6. [Additional ablations]
>
> Thank you for this suggestion. We did try using many additional models (especially mixing different model families) to compute the ranking coefficient, but that brought significant noise due to variance of evaluation results from different models. Thus the trained models underperformed the current version significantly.
>
> And following your advice, we compute the ranking coefficient based on the pythia series (14M, 31M, 70M, 160M, 410M,1B),  where we find the top domains of the llama produced data also appears in the top domains of the pythia produced data. And same for negative domains, they also share a large portion of overlap. However, due to the limited time period, we didn’t train them. We will show analyses of detailed overlap, differences and more insights about using different families of models in the revised version following your suggestions.
>
> ---
>
> Q7. [Missing comparison to [1][3][4]]
>
> We have added comparison to ScalingFilter[1] in the rebuttal (Q1). And for [3,4], we think they share the same insights with one of our baseline perplexity filtering[5]. And we will add them into related work for discussion.
>
> ---
>
> Q8. [Questions about adult content]
>
> In our observation, literature and knowledge related contents are more reflective of downstream tasks (generally have a better rank alignment) which includes essays and some adult contents. We believe filtering adult content is an orthogonal direction that could be done separately, which is not the focus of this work.  For example, we think a rule-based pre-processing step to filter adult contents out could greatly mitigate this issue.
>
> ---
>
> [5] Wenzek et.al., LREC 2020. CCNet: Extracting High Quality Monolingual Datasets from Web Crawl Data

---

> > ### Comment · Reviewer_xzzW · 2025-04-02
> >
> > Thank you for your response. I believe these additional results and discussions are valuable and make the paper much stronger and I hope you can emphasize them in the updated draft - I would even encourage you strongly to highlight this in a rewritten introduction.
> >
> > I am impressed you were able to obtain a ScalingFilter baseline so rapidly. I wonder if you could share the spearman correlation between the ScalingFilter perplexity ratios and your proposed PreSelect scores? (I mean the actual values and not the values predicted by the the fasttext models)

---

> > > ### Author Response · Authors · 2025-04-02
> > >
> > > Thank you so much for your acknowledgement for our additional results and discussions. And we appreciate your suggestions which strengthen the quality of our paper. We will definitely emphasize these in our updated draft.
> > >
> > > It does take us several days to set up the ScalingFilter baseline but since we are based on our largest model (Llama-65B) and smallest model (Llama-7B) which we have already stored the normalized loss before and training a 1B model with 30B token on a 8 * H100 node takes around one day. Those reasons ensured a timely delivery of our additional experiments and analysis.
> > >
> > > Following your suggestions, we calculate the spearman correlation between the ScalingFilter perplexity ratios and our PreSelect predictive strength, which is 0.0533. And they also have a pearson correlation of -0.079 which indicate a low correlation between these two metrics. These are measured based on the actual predictive strength score where we calculate based on the sampled subset.

---

### Official Review · Reviewer_k3Rj · 2025-03-14

**Overall Recommendation:** 3

**Summary:**

This paper explores the problem of data selection for pretraining language models. The authors propose a lightweight method that leverages predictive strength as an indicator to determine whether a document should be included in the pretraining data. To evaluate their approach, they train a group of language models of varying sizes on datasets of different scales, selected using various methods. Their findings suggest that the proposed method outperforms other data selection techniques, leading to more diverse data domains and a more balanced distribution of data length.

**Claims And Evidence:**

One key question regarding the experimental setup is why the baseline does not include a method that uses the full dataset without applying any selection. Including such a baseline could serve as a valuable reference to measure how much performance improves or declines due to data selection. This would help in assessing the effectiveness of the proposed method more comprehensively.

**Essential References Not Discussed:**

no

**Experimental Designs Or Analyses:**

The experimental designs are valid.

**Methods And Evaluation Criteria:**

Both methods and evaluation criteria make sense.

**Other Comments Or Suggestions:**

No

**Other Strengths And Weaknesses:**

### Strengths:
1. The paper is well-structured and logically organized.
2. It introduces a new method for data selection in language model pretraining.
3. The authors conduct a comprehensive analysis comparing their method to baseline approaches.
4. In addition to evaluating performance, the paper provides a detailed analysis of the characteristics of selected data and how they differ from those chosen by previous methods.

### Weaknesses
1. The experimental design does not fully verify the core assumption of the paper. The assumption is that data on which loss can better predict performance is more beneficial for pretraining. To validate this, a baseline that trains on the entire dataset without selection could be added. This would help determine the actual contribution of the unselected data and provide a clearer picture of the impact of data selection.

**Questions For Authors:**

No

**Relation To Broader Scientific Literature:**

Several prior works have explored different data selection methods for pretraining. Additionally, some studies suggest that the loss on specific data can reflect model performance on downstream benchmarks. This paper builds on that idea by assuming that data which better reflects model capability is more beneficial for pretraining. Based on this assumption, the authors introduce the predictive strength score as a criterion for selecting pretraining data.

**Theoretical Claims:**

No.

---

> ### Author Rebuttal · Authors · 2025-04-01
>
> Dear Reviewer k3Rj,
>
> Thank you for your valuable suggestions and insightful comments! We are grateful that you found our work well-structured and logically organized, and we deeply appreciate your recognition of the novelty of our method and the comprehensive analysis in our experiments.
>
> We address your comments one by one as follows:
>
> ---
>
> Q1. [Experimental design does not fully verify the core assumption]
>
> We think the most direct way to verify our core assumption is to compare the results between our selected data and randomly selected data, with the same amount of data trained. And we show this in Table 1 400M/1B/3B row Random & PreSelect, Table 2 410M/1B row Random & PreSelect, where training using data selected by PreSelect consistently and significantly outperforms random selection across model sizes and tasks.
>
> ---
>
> Q2. [Missing baseline that trains on the entire dataset without selection could be added]
>
> We thank you for highlighting this important point. In fact, on top of the random selection (i.e. without selection strategy), we did include a baseline that trains on the entire dataset (i.e. without selection). In Table 1 Row 4, we refer to it as “random with 300B”, the model is trained on the entire 300B dataset without selection, and this 300B dataset was our data pool for 1B experiments. The following rows are different selection strategies performed on this entire dataset/pool. We can see that by filtering the high-quality examples, PreSelect not only achieves a 10x reduction in compute requirements but also yields additional performance gains.
>
> ---
>
> Thank you again for your suggestions and we will describe this baseline more clearly in the revision.

---

### Official Review · Reviewer_95of · 2025-03-14

**Overall Recommendation:** 1

**Summary:**

The paper Predictive Data Selection (PRESELECT) introduces a method for selecting pretraining data based on its predictive strength, defined as the correlation between normalized loss values and downstream task performance rankings. It proposes using a fastText classifier trained on documents with high predictive strength scores to efficiently select high-quality pretraining data. The paper claims that PRESELECT outperforms other data selection methods, including PPL correlation, by offering finer granularity and improved efficiency in language model pretraining.

**Claims And Evidence:**

Normalized loss (compression efficiency) is predictive of downstream task performance. The authors cite prior work (Huang et al., 2024) showing that losses on certain text domains correlate with task performance. They extend this idea to a finer document level.
PRESELECT is more effective than existing data selection methods, including PPL correlation. Experiments demonstrate performance improvements on 17 benchmarks, outperforming PPL correlation and other baselines. PRESELECT offers a more scalable and efficient approach to data selection. The method eliminates human heuristics, operates at the document level (not just domain level), and requires only a lightweight fastText classifier.

**Essential References Not Discussed:**

The authors misrepresent PPL correlation by suggesting that it only uses Pearson correlation, despite the PPL paper pre-registering rank correlation-based experiments. They do not cite the updated version of the PPL correlation paper that includes additional benchmarks and domain-level experiments.

**Ethical Review Concerns:**

The paper titled: "Predictive Data Selection: The Data That Predicts Is the Data That Teaches" shares an alarming amount of overlap with the paper "Improving pretraining data using perplexity correlations" published at ICLR 2025 [Thrush et al., 2024]. I have been in conversations with the authors who agree and have reached out for comments. It seems there's a fundamental disagreement. I would argue that this paper is not fit for publication until this has been resolved.

**Ethical Review Flag:**

Flag this paper for an ethics review.

**Ethics Expertise Needed:**

["Research Integrity Issues (e.g., plagiarism)"]

**Experimental Designs Or Analyses:**

The experiments compare PRESELECT with multiple baselines using standardized datasets. The study measures predictive strength using rank-based correlation rather than Pearson correlation, arguing for better robustness. The claim that PRESELECT outperforms PPL correlation is supported by experimental results but does not fully acknowledge that the core methodology (perplexity-based ranking) was introduced in the PPL paper.

**Methods And Evaluation Criteria:**

The paper evaluates PRESELECT against several baselines, including: PPL correlation, DCLM and FineWeb-Edu, and random selection and low-perplexity filtering.

Performance is assessed on 17 NLP benchmarks spanning general understanding, knowledge-intensive tasks, mathematics, and coding. The evaluation criteria include: Accuracy on NLP tasks, bpc for math, and code tasks, compute efficiency.

**Other Comments Or Suggestions:**

The paper should explicitly acknowledge the PPL correlation work as a foundation and clarify what aspects are genuinely novel.
Instead of dismissing domain-level PPL correlation, the paper should compare document-level vs. domain-level ranking on the same benchmarks to fairly assess their differences.
The authors should engage with the PPL paper authors to resolve these concerns before publication.

**Other Strengths And Weaknesses:**

Strengths: efficient, scalable data selection method, evaluated on many bench marks, demonstrates compute savings.

Weaknesses: The novelty claim is questionable, given that PPL correlation pre-registered similar ideas months prior. The comparison with PPL correlation is misleading: The authors claim PPL correlation only uses Pearson correlation, which is false. They do not properly acknowledge the prior work’s pre-registration of page-level results. Overlap in experimental setup suggests that PRESELECT may be an incremental extension rather than a fundamentally new contribution.

**Questions For Authors:**

Why does the paper claim to be the first to propose rank-based correlation when the PPL correlation paper pre-registered similar results months earlier?

How does PRESELECT fundamentally differ from PPL correlation, apart from working at a document level instead of a domain level?

Why does the paper suggest that PPL correlation only uses Pearson correlation when the PPL paper explicitly discusses rank-based methods?

Given that the PPL paper now evaluates on 22 benchmarks (per the OpenReview update), does PRESELECT still maintain a clear experimental advantage?

How do the selected datasets from PRESELECT and PPL correlation compare qualitatively? Is there concrete evidence that document-level selection is superior to domain-level selection?

Would the authors be open to revising their claims to reflect PPL correlation’s prior contributions?

**Relation To Broader Scientific Literature:**

The paper references prior work on compression as intelligence (Huang et al., 2024) and perplexity-based filtering (Thrush et al., 2024).
It places PRESELECT within the context of data selection research, contrasting it with heuristic-based and supervised selection methods.
However, it does not adequately credit perplexity correlation methods in prior literature as a direct predecessor.

**Theoretical Claims:**

There's no theoretical claims in the paper, but emperic investigations of a hypothesis that ranking-based correlation is more robust than Pearson correlation for identifying predictive pretraining data. It suggests that documents where normalized losses strongly correlate with model rankings are more useful for training. This claim is similar to that of the PPL correlation paper but is framed as an original insight, despite the PPL paper pre-registering experiments on a similar premise.

---

> ### Author Rebuttal · Authors · 2025-04-01
>
> Dear Reviewer 95of,
>
> Thank you for your review. Our submission was flagged with “Research Integrity Issues (e.g., plagiarism)” due to concerns related to the PPL correlation paper.  We believe this is an unreasonable accusation and misleading to others. As we respond to the specific points below, we hope you can properly revise the statement. We are open to resolve concerns in our paper to discuss the relations between the two works more clearly (as we will mention below), but we also refute unreasonable accusations.
>
> First of all, we would like to sort out the timeline of different versions of the PPL correlation paper. The first version on arxiv was released on Sep 9, 2024 (PPLv1). The second version on arxiv was updated on March 10, 2025 (PPLv2) after the ICML submission. In our submission, we mainly referred to PPLv1. **While we will also discuss PPLv2 below, it should not be considered when judging this paper.**
>
> Second, As PPLv1 was released 5 months prior to this submission, we did not really mention our works are concurrent following the common definition of “concurrency” – **but in reality, these two works are indeed concurrent and we independently developed this idea inspired by Huang et al. 2024. PPLv1 was released early partially because it only conducted 160M-parameter experiments on small scales, and we aimed for 1B and 3B scales, more serious experimental settings (pre-filtered data pool), and more benchmarks so it took longer.** This may be the main reason why we write this paper in the current style and the reviewer feels PPLv1 was not acknowledged enough.
>
> Even so, we note that our submission and PPL correlation still admit significant differences, and lead to very different empirical results in the end. We continuously acknowledged the prior contribution of PPL correlation throughout our paper – where we included a dedicated paragraph in the Introduction section to discuss it, and made comparison with it empirically in the experiments. We address the concerns one by one below:
>
> ---
>
> Q1. [ This claim is similar to that of the PPL correlation paper but is framed as an original insight, despite the PPL paper pre-registering experiments on a similar premise.]
>
> In the introduction section, we explicitly pointed out that our submission shares similar intuition with PPLv1 (Line 95) – we were quite open acknowledging this. However, as we mentioned above, our idea was independently inspired by Huang et al. 2024 and this project started prior to release of PPLv1, thus we wrote the paper following our actual logic flow of performing this project.
>
> To mitigate the concerns on this point, we will revise our description of PPL correlation as “the first to explore domain-level correlation-based methods in data selection”.
>
> ---
>
> Q2.[The authors state they will open-source their trained data selection scorer and datasets. However, they have not done it yet]
>
> We were planning to release them after the review period.
>
> ---
>
> Q3. [Misrepresent PPL correlation by suggesting that it only uses Pearson correlation]
>
> We didn’t suggest  PPL correlation only uses Pearson correlation. The reviewer got this impression maybe due to Line 116? We never intended to imply that, Line 116 only meant that grouping documents together as in PPLv1 could enable more stable computation of Pearson correlation. We will revise that sentence to avoid misunderstanding, and in the meanwhile, we will acknowledge PPLv1’s discussion on ranking-based methods in Section 2.2.
>
> ---
>
> Q4. [They do not cite the updated version of the PPL correlation paper.]
>
> If the reviewer refers to PPLv2, we could not cite it in the submission because it was released afterwards.
>
> ---
>
> Q5. [The novelty claim is questionable, given that PPL correlation pre-registered similar ideas months prior]
>
> We do not think the community has reached agreement on how to understand or treat the  “preregistration experiments”, and it is not common practice to write them. All the conferences have no guidelines about it. While we fully understand the motivation of doing this from PPLv1, we do not consider a long list of "we will do xxxx in the future" statements to constitute prior contributions. We respectfully disagree with the notion that it is good practice for the community to simply post preliminary results and preregister experiments to claim priority.
>
> ---
>
> Because of the imposed character limitation, we keep addressing the remaining concerns in the response to reviewer VRK4. Sorry for any inconvenience and thank you for your understanding.

---

> > ### Comment · Reviewer_95of · 2025-04-01
> >
> > Five months can no longer be considered concurrent, especially given the code has been open source. You have to resolve the concerns and adequately address prior work in your paper for my score to change.

---

> > > ### Author Response · Authors · 2025-04-05
> > >
> > > Dear Reviewer 95of,
> > >
> > > To clarify, in our paper, we never mention PPLv1 was concurrent work, instead we acknowledged it in many places and empirically compared it. In the rebuttal we just wanted to explain that our idea was independently developed. Sorry for any confusion caused by the wording in our previous rebuttal.
> > >
> > > &nbsp;
> > >
> > > In our rebuttal, we believe we clearly addressed the differences between our work and PPLv1 and responded to all the reviewer’s questions. We are willing to revise some parts of the paper to address the reviewer’s concern as explained in the previous rebuttal, but we are unsure how to address the concerns further. To summarize the main points again (for reference to other reviewers and the ACs as well):
> > >
> > > 1. **We explicitly acknowledged the contributions of PPLv1 (the first version of the Perplexity Correlation paper) in multiple parts of our paper**, including a dedicated paragraph in the Introduction section to discuss it – where we acknowledged the high-level intuitions are similar, explicit statements that PPL correlation is the most relevant work (Line 169), as well as empirical comparison to it.
> > >
> > > &nbsp;
> > >
> > > 2. Compared to PPLv1, the main differences of our work are:
> > >
> > >     a. **PPLv1 adopted domain-level estimation of correlation while we use document-level**, which proves important empirically as we show in experiments.
> > >
> > >     b. **PPLv1 suggests using a diverse set of open-source LLMs to approximate correlations. In contrast, we used six Llama models, which avoids the sensitivity of different base models to evaluation configurations (e.g., prompts)**. In our preliminary experiments, using diverse open LLMs did not outperform random data selection – this aligns with PPLv2’s findings, where they failed to significantly outperform random selection with pre-filtered, high-quality data pools.
> > >
> > >     c. **PPLv1 only conducted experiments for 160M models on limited benchmarks**, while we showed effectiveness of our approach up to 3B models trained on 100B tokens and assessed on 17 benchmarks.
> > >
> > > &nbsp;
> > >
> > > 3. The reviewer seems to be curious about the comparison between our paper and PPLv2 (the latest version of perplexity correlation) as well. First, PPLv2 was released in March after the submission, thus **it should not be considered when reviewing our paper**. Second, even compared to PPLv2, the key difference noted in 2.b above remains, which leads to distinct experimental results: **With a pre-filtered data pool, PPLv2 indicates quite negative results and it does not outperform the random baseline or DCLM significantly, while we achieve pretty strong results beating all the baselines under the DCLM pre-filtered data pool.**
> > >
> > > &nbsp;
> > >
> > > 4. PPLv1 “preregistered” many experiments as future tasks, some of them overlap with our settings. However:
> > >
> > >     a. These “preregistered experiments” were not conducted at the time of our submission, even five months after they were initially proposed.
> > >
> > >     b. We do not believe that preregistered experiments diminish our contributions. The research community has not reached a consensus on whether merely preregistering experiments constitutes a valid claim of prior contributions, and we leave it to others to form their own judgments on this matter.
> > >
> > >     c. Our key difference (point 2.b above) remains and was not preregistered by PPLv1.
> > >
> > >     d. As mentioned in point 3 above, even after incorporating these preregistered experiments, PPLv2 does not perform well on pre-filtered data pools, further underscoring the uniqueness and effectiveness of our approach.
> > >
> > > &nbsp;
> > >
> > > 5. We are very willing to revise some parts of our paper to mitigate the reviewer’s concerns, which we believe can be quickly modified, including:
> > >
> > >     a. Acknowledge that PPLv1 discusses ranking-based correlation metric
> > >
> > >     b. Acknowledge preregistered experiments of PPLv1
> > >
> > >     c.Add additional baselines from Q6 in the first rebuttal

---

### Official Review · Reviewer_VRK4 · 2025-03-22

**Overall Recommendation:** 5

**Summary:**

This paper leverages the intuition that data on which losses of N models correlate with the benchmark performance (ranking) of the N models is the most useful. The paper proposes a simple numerical score for measuring this consistency in ranking (different and more numerically stable than Pearson correlation). To scale this effectively, after applying this method on a small subset of data, a fastText classifier is trained to classify text into that with high and low score using this methodology.

**Claims And Evidence:**

Yes, the main claim that the data that "predicts" is the most useful for training is empirically validated through experiments.
After the main empirical results showing how on the DCLM benchmark, this method achieves better accuracy than baselines, the paper provides a qualitative exploration of what data is actually selected by this method. We see a preference for selecting more from reliable sources such as wikipedia etc. Also, it seems that this method removes the bias that other data selection methods make towards shorter phrases (for example, perplexity based selection).

**Essential References Not Discussed:**

Essential references are all discussed.

**Experimental Designs Or Analyses:**

Yes.

**Methods And Evaluation Criteria:**

The DCLM benchmark is the most appropriate benchmark for evaluating data selection for pre-training.

**Other Comments Or Suggestions:**

N/A

**Other Strengths And Weaknesses:**

N/A.

**Questions For Authors:**

N/A

**Relation To Broader Scientific Literature:**

This is a very useful contribution in the literature of pre-training data selection. The idea of selection based on a validation set has been around for some time now, but the approach of rank correlation using a variety of open source models & distilling this ability into a fastText classifier is highly intuitive and effective as demonstrated by the results.

**Theoretical Claims:**

No theoretical claims are made.

---

> ### Author Rebuttal · Authors · 2025-04-01
>
> Dear Reviewer VRK4,
>
> Thank you for your insightful comments! And we are grateful that you found our work very useful and effective.
>
> ---
>
> ---
>
> As Reviewer 95of raises serious concerns about our paper and our response to that is probably related to other reviewers as well, due to the imposed character limitation, we have to list some responses to Reviewer 95of here, we are sorry for any inconvenience and thank you so much for the understanding.
>
> ---
>
> Q6. [The paper should compare document-level vs. domain-level ranking on the same benchmarks to fairly assess their differences]
>
> Our experiments in Table 1 already compared document-level vs. domain-level ranking on the same benchmarks in a fair setting. Following your suggestions, we train the domain-level fasttext which takes high correlation domains as positive examples while low remaining domains as negative examples, to fairly assess the difference.
>
> |**Method**|**Average Performance**|
> |:-:|:-:|
> |Random |37.2|
> |domain-level filtering|37.1|
> |domain-level fasttext| 37.5|
> |PreSelect|**40.3**|
>
>
> By using fasttext, it performs slightly better than random baseline and domain-level filtering, with 37.5% accuracy across 15 accuracy-based benchmarks, which is lower than document-level PreSelect. As we will also discuss some details in Q8 below, incorporating many domains (especially there is noise inside each domain) in fasttext training cannot lead to robust and dominant features.
>
> ---
>
> Q7. [Is there concrete evidence that document-level selection is superior to domain-level selection?]
>
> And from the top selected data ((‘fanfiction.net', 0.36%), ('opentable.com', 0.32%),('ncbi.nlm.nih.gov', 0.30%), ('alldiscountbooks.net', 0.29%), ('hindawi.com', 0.26%), there is no significant good domains upsampled. Because of the word limit in rebuttal, we will add more comparisons (e.g. Learned fasttext features, selected data distribution…) to support the advantages of document-level over domain-level.
>
> ---
>
> Q8. [How does PRESELECT fundamentally differ from PPL correlation, apart from working at a document level instead of a domain level?]
>
> As we already acknowledged in the submission, PRESELECT and PPL correlation share similar ideas, thus the high-level idea is not that “fundamentally” different, which we never claimed to be. Even so, there are significant differences:
>
> 1. PPLv1 uses domain level to classify documents while we work on finer granularity, this causes large empirical difference as we show in our original experiments and the added results in the response to Q6,7.
>
> 2. PPLv1 suggested using a large number of open-source LLMs to approximate the correlation, while we only used 6 Llama models – while we did not mention in the submission, we tried using many open LLMs in the beginning and it did not significantly outperform the random baseline, and we finally opted to models from the same family to reduce potential noises from evaluating different-family models.
>
> 3. Empirically, compared to PPLv1 which only reported 160M-scale results, our experiments cover larger scales and comprehensive benchmarks with strong results, representing significant empirical contributions as well.
>
> ---
>
> Q9. [Why does the paper suggest that PPL correlation only uses Pearson correlation when the PPL paper explicitly discusses rank-based methods?]
>
> Please refer to Q3.
>
> ---
>
> Q10. [Given that the PPL paper now evaluates on 22 benchmarks (per the OpenReview update), does PRESELECT still maintain a clear experimental advantage?]
>
> First, the new PPL paper is updated after the ICML submission deadline,, which should not be considered when reviewing our paper. Second, even compared to PPLv2, our experimental advantage is very clear:
>
> With a pre-filtered data pool, PPLv2 indicates quite negative results and it does not outperform the random baseline or DCLM significantly, while we achieve pretty strong results beating all the baselines under the DCLM pre-filtered data pool. We think that working with a pre-filtered data pool represents a more realistic setting, and empirically outperforming the baselines or not on a pre-filtered data pool makes a huge empirical difference and it directly impacts whether this method will really be used in practice or not.

---

> > ### Comment · Reviewer_VRK4 · 2025-04-02
> >
> > I continue to strongly recommend this paper for acceptance.

---

### Decision · Program_Chairs · 2025-05-01

**Decision:**

Accept (poster)

**Comment:**

This paper proposes a data selection method PRESELECT that operates on a similar intuition to previous successful work on data selection (Perplexity Correlations): that data where model loss correlates with performance are also the data most useful for training. Reviewers all agreed that the method and the experiments were sound and impressive. All reviewers were positive (xzzW did not update their score but acknowledged their concerns were addressed in the rebuttal discussion) except for one, which I discuss below.

The main negative point raised by one reviewer was severe concern about misrepresentation of the prior Perplexity Correlations work in this work. On one hand, I sympathize with the reviewers point: the authors should be careful about making statements like "we hypothesize that data on which model losses are predictive of downstream abilities also contribute effectively to learning" - even if the authors were independently inspired, it is not appropriate to frame this as a hypothesis of this work given that the hypothesis has already been posed (and verified!) by the Perplexity Correlations work.

On the other hand, I don't see good evidence that there is genuine academic misconduct here: there is no explicit plagiarism of text (as far as I can tell) nor is there evidence of the "parallel structure" between the two papers that would indicate some kind of purposeful copying. Instead, it seems as though the authors were independently inspired, found out their idea was very similar to Perplexity Correlations, and tried to differentiate their work - falling short in some respects.

As for the "truly novel" contributions of the paper, I think the combination of document-level selection, larger-scale experiments, and model curation are of general interest to the community - there is still a lot of disagreement in the broader community about the utility and scalability of data selection for pre-training, so even incremental progress here is quite interesting.

Given that the authors seem open to editing their paper, I strongly encourage them to:

1. Mention perplexity correlations in the abstract, and make the contributions of this paper more upfront. Something like "this hypothesis was first explored by [PPL]---in this work, we further corroborate and refine this hypothesis via A, B, and C."
2. Remove statements like the one quoted above (there is a similar statement in the introduction, and scattered throughout the paper).
3. More explicitly compare the design choices made to perplexity correlations in each subsection - even if the authors were independently inspired, it is important (and does not detract from the paper) to put their method into context with the current state-of-the-art.